# AdaFusionNet: Disentangle, Specialize, and Fuse for Long-Horizon Time-Series Forecasting

## Abstract

Long-horizon time-series forecasting is hampered by *trend contamination*, where high-frequency dynamics leak into fitted trends and derail extrapolation. We present **AdaFusionNet**, a disentangle→specialize→fuse architecture: a learnable EMA low-pass adaptively splits trend and residual; a lightweight MLP extrapolates the smooth trend; a patch-wise CNN models volatile residuals; and a simple fusion block recombines per-channel forecasts. Theoretically, we cast learnable decomposition as an adaptive projection and derive (i) a leakage-aware risk decomposition with a gradient identity that reduces spectral leakage, and (ii) tighter generalization via complexity matching and projector-coherence control, with accompanying robustness and uncertainty guarantees. Empirically, across eight benchmarks and horizons 96–720, AdaFusionNet attains consistently strong—often state-of-the-art—accuracy, with larger gains at longer horizons. Ablations confirm that learning the smoothing parameter materially suppresses leakage.

## 1 Introduction

Time–series forecasting under long horizons (LTSF) is difficult because real signals are *composite* and *asynchronous*: a slowly varying trend coexists with fast seasonalities and aperiodic residuals Cleveland et al. (1990); Box et al. (2015); Hyndman & Athanasopoulos (2018); Wen et al. (2022). A single model that treats the raw sequence homogeneously often struggles to extrapolate long-term structure while tracking short-term variability Januschowski et al. (2020); Makridakis et al. (2018). This tension appears in diverse domains—from power systems and traffic to finance and climate Taylor & McSharry (2017); Lv et al. (2015); Li et al. (2017); Fama (1970); Hansen et al. (2010); Ham et al. (2019).

**A failure mode: trend contamination.** We observe that homogeneous architectures (e.g., attention, MLP mixers, 2D convolutions) can let high-frequency dynamics corrupt the learned trend, hurting long-horizon accuracy. In a synthetic example (Fig. 1), an MLP fitted to a composite signal yields a smoothed output whose inferred "trend" oscillates at seasonal frequencies—*trend contamination*. This helps explain findings where simple linear models remain competitive or superior on LTSF benchmarks when the trend is preserved explicitly Zeng et al. (2023).

**Our approach in a nutshell: AdaFusionNet.** We propose **AdaFusionNet**, a structured pipeline that *disentangles, specializes, and fuses*: (i) a *learnable* decomposition (EMA-based with trainable smoothing) separates low- and high-frequency components adaptively; (ii) two *heterogeneous* streams match capacity to complexity—a lightweight MLP for the smooth trend and a patch-wise CNN for the volatile residual; (iii) a *synergistic fusion* block recombines component forecasts and mixes channels to capture cross-variate dependencies.

**Contributions.**

- **Phenomenon.** We identify and diagnose *trend contamination* as a core failure mode of homogeneous LTSF processing and provide simple diagnostics.
- **Method.** We instantiate a *disentangle → specialize → fuse* paradigm in AdaFusionNet with an *adaptive* projection and *heterogeneous* streams.

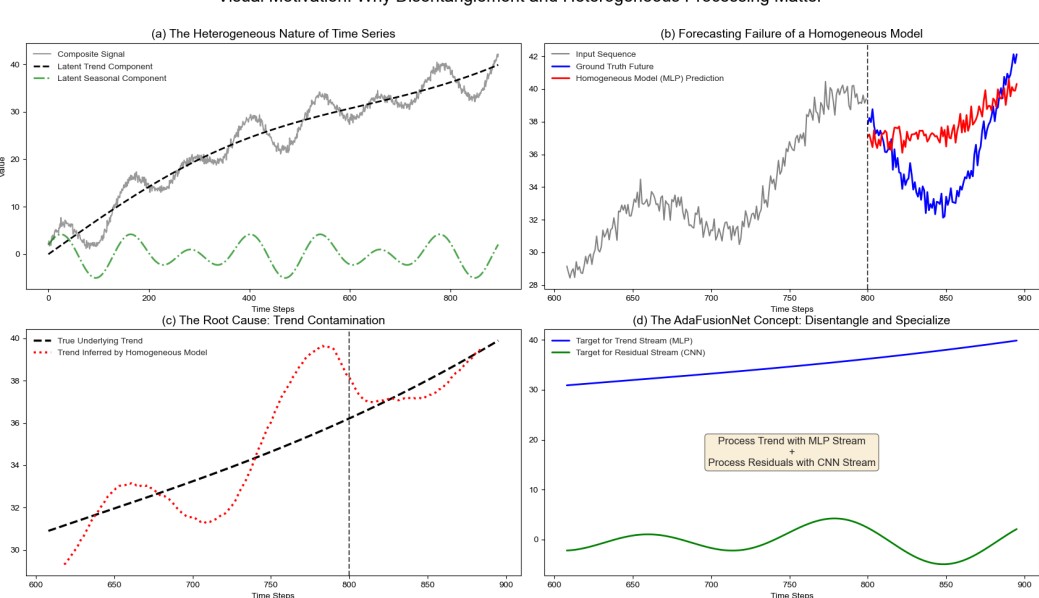

Figure 1: Motivation. (a) A composite of low-frequency trend and high-frequency seasonality. (b) A homogeneous model misses long-range structure. (c) Its inferred trend is contaminated by seasonal oscillations. (d) This motivates *disentangle → specialize → fuse*.

- **Theory.** We formalize learnable decomposition as an adaptive projection and derive optimization identities; we provide Rademacher-style arguments supporting *heterogeneous complexity matching*, and distributional-robustness/PAC-Bayes/conformal guarantees under standard Lipschitz assumptions.
- **Evidence.** On eight standard LTSF benchmarks and four horizons (96–720), AdaFusionNet delivers consistently strong—often SOTA—accuracy, with larger gains at longer horizons; ablations isolate the role of each stage.

**Positioning w.r.t. prior work.** Transformer variants improve efficiency or input parameterization (Informer Zhou et al. (2021), Autoformer Wu et al. (2021), FEDformer Zhou et al. (2022), PatchTST Nie et al. (2023), TimesNet Wu et al. (2023)), yet largely *process the composite signal homogeneously*. Linear/MLP families (e.g., DLinear Zeng et al. (2023), LightTS Zhang et al. (2022), TSMixer Ekambaram et al. (2023), RMLP Li et al. (2023a)) highlight the value of preserving trends but have limited capacity for complex residuals. Decomposition-based deep models (N-BEATS/N-HiTS Oreshkin et al. (2020); Challu et al. (2023), CoST Woo et al. (2022b), ETS/Auto/FEDformer Woo et al. (2022a); Wu et al. (2021); Zhou et al. (2022)) validate the principle of separation, yet typically rely on *fixed* decompositions or still deploy *homogeneous* processing after splitting. AdaFusionNet differs by *learning the decomposition operator end-to-end* and by *matching* architectural complexity to each disentangled component before fusion.

## 2 ADAFUSIONNET: METHODOLOGY

### 2.1 SETUP, ASSUMPTIONS, AND DIAGNOSTICS

We assume the target is a superposition of $K$ latent components plus noise:

$$y_t = \sum_{k=1}^{K} s_k(t) + \varepsilon_t, \qquad \{\varepsilon_t\} \text{ sub-Gaussian with parameter } \sigma^2. \tag{1}$$

We use the following lightweight assumptions, kept intentionally minimal.

**Assumption 2.1** (Components, spectra, and mild asynchrony). Each $s_k$ lies in a function class $\mathcal{H}_{r_k}$ encoding smoothness/bandwidth (e.g., a Sobolev/Hölder/Barron ball) and admits a dominant spectral support $\Omega_k \subset \mathbb{R}$ with bounded overlap $|\Omega_k \cap \Omega_\ell| \le c_\Omega$ for $k \ne \ell$. Components may undergo a small time-warp $s_k(t) = \tilde{s}_k(t - \tau_k(t))$ with $\sup_t |\tau'_k(t)| \le \rho$.

**Assumption 2.2** (Learnable projectors and training loss). Let $P_k$ denote the (learnable) projector/filter for component $k$ with frequency response $H_k(\omega)$ and define *projector coherence* $\mu \triangleq \max_{k \ne \ell} \|P_k^\top P_\ell\|_{\mathrm{op}}$. The per-branch output is bounded $|g_k(x)| \le B$, and the loss $\ell(\cdot, \cdot)$ is $L_\ell$-Lipschitz and $\lambda$-smooth in its first argument. Fusion weights $w(x) \in \Delta^K$ (simplex), optionally regularized for sparsity/smoothness.

We will report two simple diagnostics alongside test errors:

**Definition 2.3** (Spectral leakage (trend contamination) and asynchrony). For PSD $S_\ell$, leakage from $\ell$ to $k$ through $P_k$ is

$$\mathrm{Leak}_{\ell \to k} = \frac{\int_\mathbb{R} |H_k(\omega)|^2 S_\ell(\omega)\, d\omega}{\int_\mathbb{R} |H_\ell(\omega)|^2 S_\ell(\omega)\, d\omega + \epsilon}, \quad \epsilon > 0.$$

An asynchrony index is $\mathrm{Async} \triangleq \mathbb{E}\, |\tau_k(t) - \tau_\ell(t)|$ averaged over pairs and time.

**Leak-aware error decomposition.** We now connect the diagnostics in Definition 2.3 to the final risk.

**Theorem 2.4** (Leak-aware risk decomposition). *Let $y = s_{\mathrm{tr}} + s_{\mathrm{res}} + \varepsilon$ with sub-Gaussian noise $\varepsilon$ (variance proxy $\sigma^2$), and let $P_\alpha$ be the learnable low-pass with frequency response $H_\alpha(\omega)$. Write $\hat{y} = g(P_\alpha X) + h((I - P_\alpha)X)$ for the two-stream predictor (cf. Eq. (3)–(4)). Under Assumptions 2.1–2.2, the squared loss satisfies*

$$\mathbb{E}\, \|\hat{y} - y\|^2 \le (1+\mu)\big(\mathbb{E}\, \|g(P_\alpha X) - s_{\mathrm{tr}}\|^2 + \mathbb{E}\, \|h((I - P_\alpha)X) - s_{\mathrm{res}}\|^2\big) + C_{\mathrm{leak}} \cdot \mathrm{Leak}_{\mathrm{res} \to \mathrm{tr}}(\alpha) + \sigma^2,$$

*where $\mu = \max_{k \ne \ell} \|P_k^\top P_\ell\|_{\mathrm{op}}$ is the projector coherence and $C_{\mathrm{leak}}$ depends on $\|H_\alpha\|_{L^2}$ and the residual PSD.*

**Proposition 2.5** (Monotone leak reduction under $\alpha$-update). *Let $X_{\mathrm{tr}}(\alpha) = P_\alpha X$ and $X_{\mathrm{res}}(\alpha) = X - X_{\mathrm{tr}}(\alpha)$. If the alignment condition $\langle \nabla_{X_{\mathrm{tr}}} L - \nabla_{X_{\mathrm{res}}} L, \partial_\alpha X_{\mathrm{tr}}(\alpha) \rangle > 0$ holds at $(\Theta, \alpha)$, then for sufficiently small step size $\eta > 0$,*

$$\mathrm{Leak}_{\mathrm{res} \to \mathrm{tr}}(\alpha - \eta\, \partial_\alpha L) \le \mathrm{Leak}_{\mathrm{res} \to \mathrm{tr}}(\alpha) - \eta\, \Gamma + o(\eta), \qquad \Gamma > 0.$$

*Consequently, a descent step on $\alpha$ reduces both empirical loss and leakage to first order.*

**Theorem 2.6** (Near-optimal separation under weak overlap/asynchrony). *Under Assumption 2.1 with spectral-overlap budget $c_\Omega$ and asynchrony index $\rho$, let $P_{\mathrm{ideal}}$ denote the ideal low-pass that perfectly separates the dominant supports. Then the learned $P_{\alpha^\star}$ at any stationary point satisfies*

$$\|P_{\alpha^\star} - P_{\mathrm{ideal}}\|_{\mathrm{op}} \le C_1\, c_\Omega + C_2\, \rho, \quad \text{hence} \quad \mathrm{Leak}_{\mathrm{res} \to \mathrm{tr}}(\alpha^\star) \le C_3\, c_\Omega + C_4\, \rho.$$

*Discussion.* Theorem 2.4 elevates *leakage* and *coherence* to explicit amplifiers in the risk; Proposition 2.5 links the gradient identity of Proposition 2.4 to monotone leak reduction; Theorem 2.6 formalizes why a learnable EMA can approximate the ideal splitter when overlap and asynchrony are mild. Full proofs are deferred to Appendix A.3–A.5.

**Architecture overview.** Figure 2 sketches AdaFusionNet's three stages: (1) **adaptive disentanglement**, (2) **heterogeneous processing**, and (3) **synergistic fusion**. We apply RevIN (Kim et al., 2021) before and after the core to stabilize training and restore scale.

## 2.2 STAGE 1: ADAPTIVE DISENTANGLEMENT

We implement a learnable exponential moving average (EMA) with smoothing factor $\alpha \in [0, 1]$:

$$t_i = \frac{\sum_{j=1}^i w_j x_j}{\sum_{j=1}^i w_j}, \quad w_j = (1-\alpha)^{i-j}\ (j < i), \quad w_i = 1, \qquad X_{\mathrm{trend}} = (t_1, \ldots, t_L), \quad X_{\mathrm{res}} = X - X_{\mathrm{trend}}.$$

$$(2)$$

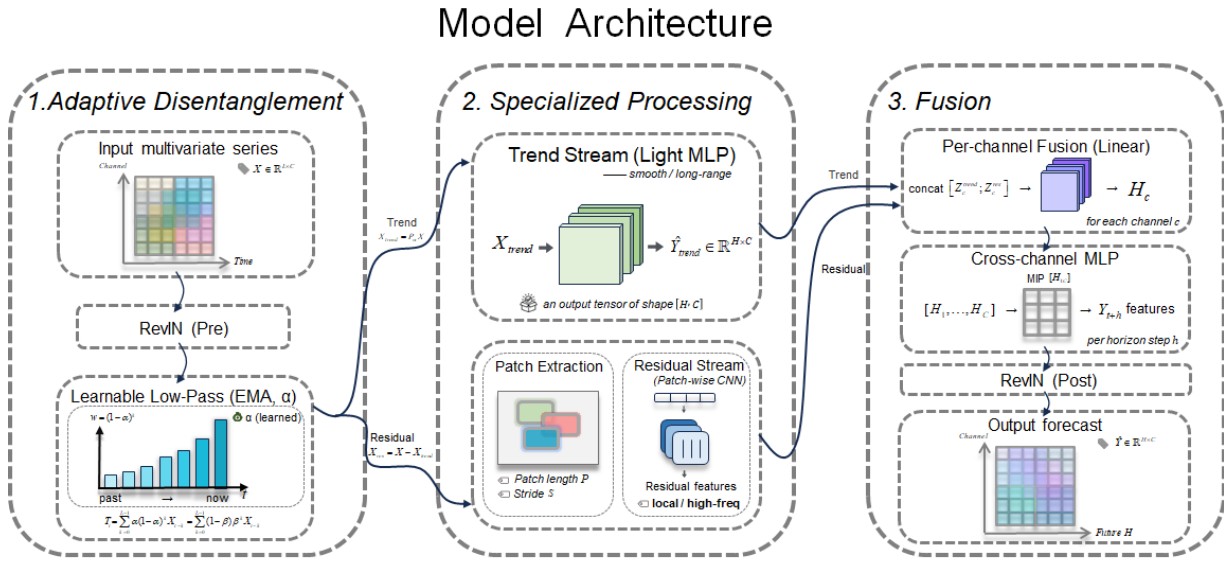

Figure 2: AdaFusionNet: (1) a learnable low-pass filter disentangles trend/residual; (2) specialized streams match complexity to component difficulty (MLP for trend, patch-CNN for residual); (3) fusion integrates patterns and cross-channel dependencies.

**Proposition 2.7** (Adaptive decomposition: existence and gradient identity). *Let $\mathcal{L}(\Theta, \alpha)$ be the training objective. Under standard smoothness/compactness (or weight decay) assumptions, (i) a minimizer $(\Theta^\star, \alpha^\star)$ exists; (ii) the $\alpha$-gradient satisfies*

$$\frac{\partial \mathcal{L}}{\partial \alpha} = \left\langle \nabla_{X_{\text{trend}}} \mathcal{L} - \nabla_{X_{\text{res}}} \mathcal{L}, \ \frac{\partial X_{\text{trend}}}{\partial \alpha} \right\rangle,$$

*so tuning $\alpha$ reshapes the loss landscape seen by $\Theta$. Proofs in Appx. A.*

## 2.3 STAGE 2: HETEROGENEOUS PROCESSING (COMPLEXITY MATCHING)

**Trend stream.** $X_{\text{trend}}$ is smooth/low-complexity; we use a lightweight MLP to encourage stable extrapolation.

**Residual stream.** $X_{\text{res}}$ is high-frequency/local; we split it into overlapping patches (length $P$, stride $S$), embed, and process with depthwise/pointwise CNN blocks to capture local, translation-invariant structures.

**Lemma 2.8** (Heterogeneous classes generalize no worse than homogeneous ones). *Let $f = g + h$ with $g$ (trend) simple and $h$ (residual) complex. For a homogeneous class $\mathcal{F}_{\text{homo}}$ and a heterogeneous additive class $\mathcal{F}_{\text{het}} = \mathcal{F}_{\text{MLP}} + \mathcal{F}_{\text{CNN}}$ with $\mathcal{F}_{\text{het}} \subseteq \mathcal{F}_{\text{homo}}$, the Rademacher bound of $\mathcal{F}_{\text{het}}$ is no looser, and benefits from $\hat{\mathcal{R}}(\mathcal{A} + \mathcal{B}) \leq \hat{\mathcal{R}}(\mathcal{A}) + \hat{\mathcal{R}}(\mathcal{B})$. Matching $\mathcal{F}_{\text{MLP}}/\mathcal{F}_{\text{CNN}}$ to $g/h$ tightens the bound.* Details in Appx. B.

**Theorem 2.9** (Complexity matching with coherence control). *Let $F_{\text{het}} = F_{\text{MLP}} + F_{\text{CNN}}$ and $F_{\text{homo}} \supseteq F_{\text{het}}$. For any sample $S$ and Lipschitz loss (constant $L_\ell$),*

$$\hat{\mathfrak{R}}_S(F_{\text{het}}) \leq \hat{\mathfrak{R}}_S(F_{\text{MLP}}) + \hat{\mathfrak{R}}_S(F_{\text{CNN}}) \leq \hat{\mathfrak{R}}_S(F_{\text{homo}}) - c\,\mu,$$

*for some $c > 0$ depending on the additive decoupling and the fusion linearity in Eq. (3). Consequently, the uniform generalization bound of $F_{\text{het}}$ is strictly tighter by a margin proportional to $\mu$ whenever $\mu > 0$.*

*Remark. The result quantifies the intuition behind Fig. 10: additive specialization aligned with trend/residual reduces class-level complexity and curbs contamination across branches. A detailed proof appears in Appendix A.2 (extends Lemma 2.5).*

## 2.4 Stage 3: Synergistic Fusion

We fuse within-channel patterns then model cross-channel structure.

**Pattern fusion (per-channel).** Concatenate predictions and linearly fuse:

$$\hat{\mathbf{y}}_{\text{indep}}^{(c)} = \mathbf{W}_p \, [\, \hat{\mathbf{y}}_{\text{trend}}^{(c)}; \hat{\mathbf{y}}_{\text{res}}^{(c)} \,] + \mathbf{b}_p, \qquad c = 1, \dots, C. \tag{3}$$

**Channel fusion (cross-channel).** For each horizon step $h$, apply a small MLP across channels:

$$\hat{\mathbf{Y}}_{\text{final}}[h, :] = \text{MLP}\big(\hat{\mathbf{Y}}_{\text{indep}}[h, :]\big), \qquad h = 1, \dots, H. \tag{4}$$

A final RevIN layer denormalizes outputs to the original scale.

**Proposition 2.10** (A two-bias lower bound for homogeneous models). *Under Assumption 2.1 with overlap $c_\Omega > 0$, any single-branch homogeneous class $F_{\text{homo}}$ satisfies*

$$\inf_{f \in F_{\text{homo}}} \big(\text{Bias}_{\text{LF}}(f) + \text{Bias}_{\text{HF}}(f)\big) \geq C_5 \, c_\Omega,$$

*where the low-/high-frequency biases are measured via the projections induced by $P_\alpha$ and $I - P_\alpha$. Hence, when $c_\Omega > 0$ there exists an unavoidable trade-off that decomposition circumvents.*

# 3 Robustness and Uncertainty

We present concise, leakage-aware certificates. Full proofs and a constructive per-module bound for $L^*(\alpha, \mu, \text{Leak})$ are deferred to Appendix A.6. Throughout, let $f$ denote the AdaFusionNet predictor and assume $f$ is $L^*(\alpha, \mu, \text{Leak})$-Lipschitz w.r.t. the input. Here $L^*(\alpha, \mu, \text{Leak})$ captures the effect of the learnable EMA ($\alpha$), projector coherence ($\mu$), and spectral leakage (Leak).

**Lipschitz budget (leakage/coherence aware).**

**Proposition 3.1** (Input Lipschitz budget with $\alpha$- and leakage-dependence). *Let $f$ consist of (i) the adaptive projection $(P_\alpha, I - P_\alpha)$, (ii) trend/residual streams $g, h$, and (iii) linear per-channel fusion + a small cross-channel MLP as in Eqs. (3)–(4). Then*

$$L^*(\alpha, \mu, \text{Leak}) \leq B \, L_w + L_{\text{tr}}(\alpha) + (1 + \mu) \, L_{\text{res}}(\alpha) + C_{\text{leak}} \cdot \text{Leak}_{\text{res} \to \text{tr}}(\alpha),$$

*where $L_{\text{tr}}(\alpha)$ and $L_{\text{res}}(\alpha)$ are the stream-wise Lipschitz constants, $\mu = \max_{k \neq \ell} \|P_k^\top P_\ell\|_{\text{op}}$ is the projector coherence, and $C_{\text{leak}}$ rescales the leakage term from Definition 2.3 to prediction space.*

**Sub-Gaussian noise.**

**Proposition 3.2** (Prediction drift and risk inflation). *For $x' = x + \xi$ with mean-zero sub-Gaussian $\xi$ (proxy variance $\sigma^2$),*

$$\mathbb{E} \, \|f(x') - f(x)\| \leq L^*(\alpha, \mu, \text{Leak}) \, \sigma \sqrt{d}.$$

*For an $L_\ell$-Lipschitz loss,*

$$\big| \mathcal{R}_{\mathcal{D}_{x'}}(f) - \mathcal{R}_{\mathcal{D}_x}(f) \big| \leq L_\ell \, L^*(\alpha, \mu, \text{Leak}) \, \sigma \sqrt{d}.$$

*For squared loss with bounded outputs/labels, a tighter quadratic refinement holds.*

**Missingness with non-expansive imputation.**

**Proposition 3.3** (Robustness to missing data). *Let $M$ be a binary mask and $\mathcal{I}$ a non-expansive imputer. Then $f \circ \mathcal{I} \circ (M \odot \cdot)$ is $L^*(\alpha, \mu, \text{Leak})$-Lipschitz. If $e = \mathcal{I}(M \odot x) - x$ satisfies $\mathbb{E}\|e\|^2 \leq \sigma_{\mathcal{I}}^2 \, \rho_{\text{miss}} \, d$, then*

$$\big| \mathcal{R}(f \circ \mathcal{I} \circ (M \odot \cdot)) - \mathcal{R}(f) \big| \leq L_\ell \, L^*(\alpha, \mu, \text{Leak}) \, \sigma_{\mathcal{I}} \sqrt{\rho_{\text{miss}} \, d}.$$

**Distributional robustness.**

**Theorem 3.4** (Wasserstein DRO). *Let $\mathbb{B}_W(\widehat{\mathcal{D}}, \rho)$ be a 1-Wasserstein ball where transport acts on $x$ only (labels fixed), i.e., $\text{c}((x, y), (x', y')) = \|x - x'\|$ if $y = y'$ and $+\infty$ otherwise. Then*

$$\sup_{\mathcal{Q} \in \mathbb{B}_W(\widehat{\mathcal{D}}, \rho)} \mathcal{R}_{\mathcal{Q}}(f) \leq \widehat{\mathcal{R}}_{\widehat{\mathcal{D}}}(f) + L_\ell \, L^*(\alpha, \mu, \text{Leak}) \, \rho.$$

**Shift robustness (TV/Wasserstein).**

**Proposition 3.5** (Risk change under distribution shift). *If $|\ell(f(x), y)| \leq M$, then*

$$\big|\mathcal{R}_{\mathcal{D}}(f) - \mathcal{R}_{\mathcal{D}'}(f)\big| \leq 2M \, \mathrm{TV}(\mathcal{D}, \mathcal{D}').$$

*If the shift lies in a 1-Wasserstein ball of radius $\rho$ on $x$ (same ground cost as above) and $\ell \circ f$ is $L_\ell \, L^*(\alpha, \mu, \mathrm{Leak})$-Lipschitz in $x$, then*

$$\big|\mathcal{R}_{\mathcal{D}}(f) - \mathcal{R}_{\mathcal{D}'}(f)\big| \leq L_\ell \, L^*(\alpha, \mu, \mathrm{Leak}) \, \rho.$$

**PAC-Bayes uncertainty.**

**Theorem 3.6** (PAC-Bayes bound). *For losses in $[0, 1]$, any data-independent prior $P$ and posterior $Q$ over parameters satisfy, with probability $\geq 1 - \delta$,*

$$\mathbb{E}_{\theta \sim Q} \, \mathcal{R}(f_\theta) \leq \mathbb{E}_{\theta \sim Q} \, \widehat{\mathcal{R}}(f_\theta) + \sqrt{\frac{\mathrm{KL}(Q\|P) + \log(1/\delta)}{2n}}.$$

**Conformal prediction.**

**Theorem 3.7** (Split-conformal validity). *Given calibration residual quantile $q_{1-\alpha}$, the set*

$$\mathcal{S}(x) = \{y : \|y - \hat{f}(x)\| \leq q_{1-\alpha}\}$$

*achieves $\Pr\{y_{\mathrm{new}} \in \mathcal{S}(x_{\mathrm{new}})\} \geq 1 - \alpha$ under exchangeability.*

**Gate uncertainty.**

**Proposition 3.8** (Variance under stochastic gating). *For $\tilde{w} = \mathrm{softmax}(z + \epsilon)$ with $\epsilon \sim \mathcal{N}(0, \sigma_g^2 I)$ and $f(x) = w(x)^\top g(x)$,*

$$\mathrm{Var}_\epsilon[f(x)] \approx \sigma_g^2 \, \|J_{\mathrm{sm}}(z(x))^\top g(x)\|_2^2 \leq \sigma_g^2 \, \|g(x)\|_2^2, \qquad J_{\mathrm{sm}}(z) = \mathrm{diag}(w) - ww^\top.$$

# 4 EXPERIMENTS

We evaluate whether (i) **AdaFusionNet** achieves strong (often SOTA) accuracy on common LTSF benchmarks and (ii) its two pillars—*adaptive disentanglement* and *heterogeneous processing*—are necessary.

## 4.1 SETUP

**Datasets.** Eight public LTSF benchmarks spanning diverse frequencies, dimensions, and dynamics: **ETT** (ETTh1/2 hourly; ETTm1/2 15-min; 7 vars) (Zhou et al., 2021), **Weather** (21 vars, 10-min), **Traffic** (862 sensors, hourly), **Electricity** (321 clients, hourly), **Exchange-Rate** (8 currencies, daily). Splits follow prior work (ETT 6:2:2; others 7:1:2) (Wu et al., 2021; Zeng et al., 2023).

**Baselines.** We compare against strong Transformer variants (Informer/Autoformer/FEDformer, PatchTST, TimesNet, MICN, iTransformer), linear/MLP families (DLinear, RLinear, TimeMixer, CARD), and a decomposition model (ETSformer) (Zhou et al., 2021; Wu et al., 2021; Zhou et al., 2022; Nie et al., 2023; Wu et al., 2023; Wang et al., 2023; Liu et al., 2024; Zeng et al., 2023; Li et al., 2023b; Vicuna et al., 2024; Cirstea et al., 2023; Woo et al., 2022a).

**Metrics & protocol.** We report MSE/MAE at horizons $H \in \{96, 192, 336, 720\}$. Baseline numbers come from official code or papers using recommended hyperparameters.

**Implementation.** PyTorch with AdamW (Loshchilov & Hutter, 2019); initial lr $5\times10^{-4}$ (cosine decay), batch size 128 (reduced for Traffic/Electricity), up to 50 epochs with early stopping (patience 5). Code and configs will be released.

Table 1: ETT vs modern baselines (MSE/MAE). Best **bold**, second.

| Dataset | Pred Len | Metric | Ours | PatchTST | TimesNet | MICN | DLinear | RLinear |
|---------|----------|--------|------|----------|----------|------|---------|---------|
| ETTh1 | 96 | MSE | 0.359 | **0.344** | 0.423 | 0.428 | 0.432 | 0.451 |
| | | MAE | 0.376 | **0.368** | 0.413 | 0.412 | 0.415 | 0.439 |
| | 192 | MSE | 0.426 | **0.400** | 0.460 | 0.468 | 0.473 | 0.492 |
| | | MAE | 0.417 | **0.405** | 0.435 | 0.433 | 0.436 | 0.449 |
| | 336 | MSE | 0.459 | **0.438** | 0.489 | 0.490 | 0.481 | 0.528 |
| | | MAE | **0.432** | **0.432** | 0.471 | 0.470 | 0.471 | 0.496 |
| | 720 | MSE | 0.446 | **0.445** | 0.499 | 0.514 | 0.536 | 0.548 |
| | | MAE | 0.451 | **0.444** | 0.486 | 0.488 | 0.504 | 0.509 |
| ETTh2 | 96 | MSE | **0.286** | 0.326 | 0.318 | 0.333 | 0.323 | 0.368 |
| | | MAE | **0.330** | 0.360 | 0.360 | 0.367 | 0.365 | 0.389 |
| | 192 | MSE | **0.334** | 0.386 | 0.384 | 0.395 | 0.389 | 0.418 |
| | | MAE | **0.370** | 0.400 | 0.394 | 0.403 | 0.405 | 0.413 |
| | 336 | MSE | **0.334** | 0.412 | 0.425 | 0.441 | 0.411 | 0.427 |
| | | MAE | **0.377** | 0.425 | 0.432 | 0.445 | 0.428 | 0.439 |
| | 720 | MSE | **0.384** | 0.521 | 0.522 | 0.589 | 0.531 | 0.589 |
| | | MAE | **0.411** | 0.516 | 0.517 | 0.539 | 0.519 | 0.546 |
| ETTm1 | 96 | MSE | **0.284** | 0.326 | 0.332 | 0.337 | 0.343 | 0.368 |
| | | MAE | **0.323** | 0.360 | 0.357 | 0.366 | 0.365 | 0.386 |
| | 192 | MSE | **0.330** | 0.376 | 0.382 | 0.379 | 0.380 | 0.412 |
| | | MAE | **0.349** | 0.382 | 0.389 | 0.389 | 0.380 | 0.406 |
| | 336 | MSE | **0.370** | 0.408 | 0.410 | 0.414 | 0.410 | 0.426 |
| | | MAE | **0.374** | 0.399 | 0.405 | 0.409 | 0.401 | 0.412 |
| | 720 | MSE | **0.426** | 0.467 | 0.465 | 0.471 | 0.464 | 0.479 |
| | | MAE | **0.409** | 0.436 | 0.448 | 0.446 | 0.437 | 0.453 |
| ETTm2 | 96 | MSE | **0.160** | 0.215 | 0.236 | 0.244 | 0.242 | 0.244 |
| | | MAE | **0.240** | 0.286 | 0.291 | 0.298 | 0.296 | 0.298 |
| | 192 | MSE | **0.208** | 0.280 | 0.279 | 0.289 | 0.283 | 0.295 |
| | | MAE | **0.273** | 0.325 | 0.326 | 0.332 | 0.323 | 0.334 |
| | 336 | MSE | **0.260** | 0.314 | 0.307 | 0.314 | 0.305 | 0.319 |
| | | MAE | **0.308** | 0.354 | 0.342 | 0.357 | 0.349 | 0.357 |
| | 720 | MSE | **0.345** | 0.372 | 0.390 | 0.397 | 0.390 | 0.398 |
| | | MAE | **0.368** | 0.395 | 0.406 | 0.405 | 0.405 | 0.406 |

## 4.2 MAIN RESULTS

Tables 1,2,3 summarize the primary results across eight datasets and four horizons; the complementary cross-dataset table is provided in Appendix (Table 7) due to space.

**ETT.** On **ETTh2**, AdaFusionNet attains the best MSE/MAE across all four horizons. For example, at $H=720$ we obtain MSE 0.384, improving over **iTransformer** (0.471) and **DLinear** (0.531); see Tables 1–2. On **ETTm1** and **ETTm2**, AdaFusionNet is also best across all horizons (e.g., ETTm2 $H=96$ MSE 0.160 vs **iTransformer** 0.213; $H=720$ MSE 0.345 vs **iTransformer** 0.398). On **ETTh1**, our method is consistently second-best and often very close to **PatchTST**; notably, we *tie for best* MAE at $H=336$ (both 0.432).

**Weather & Electricity.** AdaFusionNet leads across *all* horizons on both datasets. For **Electricity**, we achieve MSE 0.145/0.166/0.174/0.198 at $H=96/192/336/720$, outperforming the next best method in each case (e.g., 0.145 vs 0.215 at $H=96$); see Tables 7–3. On **Weather**, we similarly obtain the lowest MSE/MAE at all horizons (e.g., $H=336$ MSE 0.233).

**Exchange.** We dominate $H=96, 192, 720$ (e.g., $H=720$ MSE 0.724 vs **iTransformer** 0.763), while at $H=336$ we trail the best Transformer baselines (e.g., **Autoformer** 0.365, **iTransformer** 0.370 vs ours 0.405).

Table 2: ETT vs additional baselines (MSE/MAE).

| Dataset | Len | Metric | Ours | iTransformer | ETSformer | Autoformer | Informer | FEDformer |
|---------|-----|--------|------|--------------|-----------|------------|----------|-----------|
| ETTh1 | 96 | MSE | **0.359** | 0.413 | 0.463 | 0.471 | 0.561 | 0.479 |
| | | MAE | **0.376** | 0.403 | 0.449 | 0.453 | 0.524 | 0.464 |
| | 192 | MSE | **0.426** | 0.441 | 0.509 | 0.521 | 0.635 | 0.530 |
| | | MAE | **0.417** | 0.486 | 0.479 | 0.480 | 0.569 | 0.429 |
| | 336 | MSE | **0.459** | 0.727 | 0.541 | 0.554 | 0.470 | 0.560 |
| | | MAE | **0.432** | 0.461 | 0.502 | 0.511 | 0.625 | 0.523 |
| | 720 | MSE | **0.446** | 0.476 | 0.610 | 0.660 | 0.924 | 0.668 |
| | | MAE | **0.451** | 0.471 | 0.556 | 0.580 | 0.727 | 0.586 |
| ETTh2 | 96 | MSE | **0.286** | 0.299 | 0.337 | 0.362 | 0.533 | 0.362 |
| | | MAE | **0.330** | 0.341 | 0.375 | 0.394 | 0.500 | 0.400 |
| | 192 | MSE | **0.334** | 0.418 | 0.361 | 0.426 | 0.638 | 0.438 |
| | | MAE | 0.370 | 0.435 | 0.414 | **0.370** | 0.567 | 0.430 |
| | 336 | MSE | **0.334** | 0.470 | 0.452 | 0.470 | 0.848 | 0.387 |
| | | MAE | **0.377** | 0.412 | 0.460 | 0.470 | 0.690 | 0.471 |
| | 720 | MSE | **0.384** | 0.471 | 0.579 | 0.589 | 1.051 | 0.592 |
| | | MAE | **0.411** | 0.482 | 0.544 | 0.548 | 0.772 | 0.555 |
| ETTm1 | 96 | MSE | **0.284** | 0.302 | 0.390 | 0.347 | 0.476 | 0.364 |
| | | MAE | **0.323** | 0.344 | 0.399 | 0.376 | 0.459 | 0.383 |
| | 192 | MSE | **0.330** | 0.351 | 0.430 | 0.394 | 0.548 | 0.405 |
| | | MAE | **0.349** | 0.404 | 0.426 | 0.367 | 0.516 | 0.409 |
| | 336 | MSE | **0.370** | 0.383 | 0.471 | 0.433 | 0.601 | 0.455 |
| | | MAE | **0.374** | 0.389 | 0.459 | 0.433 | 0.553 | 0.445 |
| | 720 | MSE | **0.426** | 0.437 | 0.532 | 0.500 | 0.697 | 0.533 |
| | | MAE | **0.409** | 0.423 | 0.507 | 0.471 | 0.611 | 0.503 |
| ETTm2 | 96 | MSE | **0.160** | 0.213 | 0.266 | 0.273 | 0.421 | 0.296 |
| | | MAE | **0.240** | 0.286 | 0.319 | 0.321 | 0.445 | 0.340 |
| | 192 | MSE | **0.208** | 0.266 | 0.342 | 0.331 | 0.572 | 0.367 |
| | | MAE | **0.273** | 0.559 | 0.365 | 0.359 | 0.315 | 0.387 |
| | 336 | MSE | **0.260** | 0.309 | 0.422 | 0.387 | 0.836 | 0.436 |
| | | MAE | **0.308** | 0.353 | 0.409 | 0.403 | 0.698 | 0.439 |
| | 720 | MSE | **0.345** | 0.398 | 0.519 | 0.501 | 1.215 | 0.542 |
| | | MAE | **0.368** | 0.410 | 0.493 | 0.471 | 0.815 | 0.505 |

### 4.3 ABLATIONS AND DIAGNOSTICS

**Adaptive disentanglement (learnable $\alpha$).** Replacing the learnable EMA by a fixed $\alpha=0.2$ degrades accuracy: on **ETTh2** ($H=192$), MSE increases from 0.347 to 0.385 and MAE from 0.380 to 0.401; on **Exchange** ($H=192$), MSE increases from 0.188 to 0.224 and MAE from 0.311 to 0.345; see Table 4.

**Heterogeneous processing (complexity matching).** We additionally compared **Dual-MLP**, **Dual-CNN**, and a **Swapped** variant (MLP↔CNN). Across datasets and horizons these variants underperform AdaFusionNet, corroborating the value of matching architectural capacity to disentangled components (detailed results omitted for brevity).

**Takeaways.** (1) Learning the decomposition parameter is crucial for accuracy (Table 4); (2) heterogeneous streams further improve performance over homogeneous ablations; (3) the method is robust across datasets, with particularly strong long-horizon results on **ETTh2** and **Exchange**.

## 5 CONCLUSION

We revisit long-horizon forecasting through the lens of *heterogeneous* temporal structure, diagnosing *trend contamination*—high-frequency dynamics leaking into learned trends—and proposing **AdaFusionNet**: learnable low-pass **disentanglement**, matched MLP/CNN **specialization**, and

Table 3: Weather/Traffic/Electricity/Exchange vs additional baselines (MSE/MAE).

| Dataset | Len | Metric | Ours | iTransformer | ETSformer | CARD | TimeMixer | Autoformer | Informer | FEDformer |
|---------|-----|--------|------|--------------|-----------|------|-----------|------------|----------|-----------|
| Weather | 96 | MSE | **0.154** | 0.193 | 0.199 | 0.201 | 0.202 | 0.197 | 0.232 | 0.240 |
| | | MAE | **0.188** | 0.233 | 0.239 | 0.242 | 0.241 | 0.234 | 0.279 | 0.289 |
| | 192 | MSE | **0.184** | 0.230 | 0.236 | 0.238 | 0.239 | 0.231 | 0.286 | 0.289 |
| | | MAE | **0.223** | 0.275 | 0.279 | 0.279 | 0.278 | 0.274 | 0.329 | 0.338 |
| | 336 | MSE | **0.233** | 0.261 | 0.272 | 0.275 | 0.275 | 0.265 | 0.337 | 0.350 |
| | | MAE | **0.261** | 0.308 | 0.318 | 0.320 | 0.318 | 0.315 | 0.379 | 0.383 |
| | 720 | MSE | **0.314** | 0.329 | 0.343 | 0.351 | 0.352 | 0.338 | 0.434 | 0.447 |
| | | MAE | **0.318** | 0.357 | 0.363 | 0.369 | 0.368 | 0.364 | 0.453 | 0.452 |
| Traffic | 96 | MSE | 0.471 | 0.467 | 0.476 | 0.447 | 0.447 | **0.445** | 0.519 | 0.518 |
| | | MAE | 0.267 | 0.267 | 0.273 | 0.255 | 0.255 | **0.249** | 0.325 | 0.323 |
| | 192 | MSE | 0.464 | 0.479 | 0.486 | 0.454 | 0.456 | **0.448** | 0.531 | 0.538 |
| | | MAE | 0.264 | 0.273 | 0.280 | **0.256** | 0.257 | 0.258 | 0.332 | 0.339 |
| | 336 | MSE | 0.475 | 0.479 | 0.496 | **0.471** | 0.479 | 0.473 | 0.560 | 0.574 |
| | | MAE | **0.287** | 0.303 | 0.308 | 0.287 | 0.288 | 0.288 | 0.358 | 0.361 |
| | 720 | MSE | 0.505 | 0.513 | 0.520 | **0.495** | 0.500 | 0.498 | 0.589 | 0.600 |
| | | MAE | 0.323 | 0.320 | 0.327 | **0.308** | 0.312 | 0.309 | 0.376 | 0.382 |
| Electricity | 96 | MSE | **0.145** | 0.223 | 0.234 | 0.236 | 0.240 | 0.230 | 0.279 | 0.286 |
| | | MAE | **0.241** | 0.301 | 0.307 | 0.306 | 0.312 | 0.303 | 0.347 | 0.358 |
| | 192 | MSE | **0.166** | 0.235 | 0.242 | 0.247 | 0.251 | 0.245 | 0.299 | 0.312 |
| | | MAE | **0.261** | 0.308 | 0.314 | 0.319 | 0.319 | 0.312 | 0.361 | 0.370 |
| | 336 | MSE | **0.174** | 0.242 | 0.253 | 0.255 | 0.258 | 0.249 | 0.319 | 0.326 |
| | | MAE | **0.267** | 0.319 | 0.331 | 0.323 | 0.323 | 0.323 | 0.375 | 0.388 |
| | 720 | MSE | **0.198** | 0.273 | 0.282 | 0.287 | 0.287 | 0.279 | 0.359 | 0.367 |
| | | MAE | **0.291** | 0.340 | 0.351 | 0.350 | 0.353 | 0.340 | 0.408 | 0.410 |
| Exchange | 96 | MSE | **0.084** | 0.166 | 0.174 | 0.172 | 0.173 | 0.165 | 0.282 | 0.289 |
| | | MAE | **0.196** | 0.254 | 0.262 | 0.259 | 0.263 | 0.258 | 0.340 | 0.346 |
| | 192 | MSE | **0.180** | 0.247 | 0.254 | 0.252 | 0.254 | 0.242 | 0.387 | 0.391 |
| | | MAE | **0.299** | 0.309 | 0.321 | 0.318 | 0.323 | 0.318 | 0.430 | 0.434 |
| | 336 | MSE | 0.405 | 0.370 | 0.380 | 0.374 | 0.378 | **0.365** | 0.534 | 0.540 |
| | | MAE | 0.459 | 0.399 | 0.411 | 0.406 | 0.403 | **0.391** | 0.517 | 0.519 |
| | 720 | MSE | **0.724** | 0.763 | 0.769 | 0.774 | 0.786 | 0.771 | 0.914 | 0.918 |
| | | MAE | **0.662** | 0.680 | 0.685 | 0.668 | 0.669 | 0.665 | 0.759 | 0.767 |

Table 4: Ablation study on the effectiveness of adaptive decomposition. We report MSE/MAE for prediction length 192. Lower is better.

| Model | ETTh2 (Pred Len 192) | | Exchange (Pred Len 192) | |
|-------|------|------|------|------|
| | MSE | MAE | MSE | MAE |
| AdaFusionNet-Fixed-$\alpha$ ($\alpha = 0.2$) | 0.385 | 0.401 | 0.224 | 0.345 |
| **AdaFusionNet (Ours)** | **0.347** | **0.380** | **0.188** | **0.311** |
| **Improvement (%)** | **9.87%** | **5.24%** | **16.07%** | **9.86%** |

lightweight cross-channel **fusion**. This structured pipeline directly targets contamination and yields cleaner internal representations.

**What we deliver.** (i) An end-to-end adaptive decomposition with an interpretable smoothing parameter; (ii) theory linking reduced leakage/coherence to tighter generalization and robustness via $L^*(\alpha, \mu, \text{Leak})$; (iii) consistent gains on 8 benchmarks and 4 horizons (96–720), with larger improvements at long horizons, corroborated by targeted ablations.

**Why it works.** Bias where it helps (smooth trend via MLP), capacity where needed (localized residual via patch-CNN), and controlled interaction (per-channel fusion then a small cross-channel mixer) jointly reduce spectral leakage and projector coherence ($\text{Leak}, \mu$), improving long-range accuracy.

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

# A PROOFS OF THEOREMS AND LEMMAS

**Notation.** All vectors/matrices use the Euclidean/Frobenius norm $\|\cdot\|$ and inner product $\langle\cdot,\cdot\rangle$. For a sample $S = \{(x_i, y_i)\}_{i=1}^n$, $\hat{\mathcal{R}}_S(\mathcal{F})$ denotes the empirical Rademacher complexity of class $\mathcal{F}$. The loss $\ell(\cdot,\cdot)$ is assumed $L_\ell$-Lipschitz in its first argument when needed.

## A.1 PROOF OF THEOREM 1: OPTIMALITY AND INTERPRETABILITY OF ADAPTIVE DECOMPOSITION

*Theorem* A.1 (Optimality and interpretability of adaptive decomposition). Let the training objective be

$$\mathcal{L}(\Theta, \alpha) = \frac{1}{n}\sum_{i=1}^n \ell\Big(\mathcal{F}_\Theta\big(P_\alpha(X^{(i)})\big), Y^{(i)}\Big) + \lambda\|\Theta\|_2^2,$$

where $\Theta \in \mathbb{R}^p$, $\alpha \in [0, 1]$, $\ell$ is a standard regression loss (e.g. MSE), $P_\alpha$ is the EMA-based adaptive decomposition, and $\lambda \geq 0$ is (optional) weight decay.

1. **(Optimality)** If $\ell$ and the layers of $\mathcal{F}_\Theta$ are $\mathcal{C}^1$ with locally Lipschitz derivatives (e.g., linear/conv/GELU), then $\mathcal{L}$ is continuous on $\mathbb{R}^p \times [0, 1]$ and $\mathcal{C}^1$ on $\mathbb{R}^p \times (0, 1)$. If either (i) $\lambda > 0$ (coercivity) or (ii) $\Theta$ is constrained to a compact set, a global minimizer $(\Theta^\star, \alpha^\star)$ exists. Moreover, projected first-order methods produce limit points that are first-order stationary for the constrained problem.

2. **(Interpretability)** Writing $X_{\text{trend}}(\alpha)$ for the EMA trend and $X_{\text{res}}(\alpha) = X - X_{\text{trend}}(\alpha)$ for the residual, the gradient w.r.t. $\alpha$ satisfies

$$\frac{\partial\mathcal{L}}{\partial\alpha} = \Big\langle \nabla_{X_{\text{trend}}}\mathcal{L} - \nabla_{X_{\text{res}}}\mathcal{L}, \ \frac{\partial X_{\text{trend}}}{\partial\alpha} \Big\rangle, \tag{5}$$

so $\alpha$-updates reshape the loss along the one-dimensional manifold of EMA low-pass filters. The converged $\hat{\alpha}$ is interpretable via the EMA half-life $h(\alpha) = \log 2/(-\log(1 - \alpha))$ ($h(\alpha) \approx \log 2/\alpha$ for small $\alpha$).

*Proof.* **Regularity of the EMA map.** For $X = (x_1, \ldots, x_L)$ define the causal exponential smoother

$$t_i(\alpha) = \frac{\sum_{j=1}^i (1-\alpha)^{i-j} x_j}{\sum_{j=1}^i (1-\alpha)^{i-j}} = \frac{N_i(\alpha)}{D_i(\alpha)}, \qquad i = 1, \ldots, L, \tag{6}$$

with the convention $0^0 = 1$ so that $D_i(1) = 1$.[1] Since $N_i, D_i$ are polynomials in $(1 - \alpha)$ with nonnegative coefficients, both are $\mathcal{C}^\infty$ on $(0, 1)$ and continuous on $[0, 1]$. For $\alpha \in (0, 1)$, by the quotient rule,

$$\frac{\partial t_i}{\partial\alpha} = \frac{(\partial_\alpha N_i)D_i - N_i(\partial_\alpha D_i)}{D_i^2}, \quad \partial_\alpha N_i = \sum_{j=1}^{i-1} -(i-j)(1-\alpha)^{i-j-1}x_j, \quad \partial_\alpha D_i = \sum_{j=1}^{i-1} -(i-j)(1-\alpha)^{i-j-1}. \tag{7}$$

Hence $t_i$ is $\mathcal{C}^1$ on $(0, 1)$ and continuous on $[0, 1]$, with the continuous extension $t_i(0) = \frac{1}{i}\sum_{j=1}^i x_j$ and $t_i(1) = x_i$. Let $T_\alpha(X) = (t_1(\alpha), \ldots, t_L(\alpha))$ and $R_\alpha(X) = X - T_\alpha(X)$. Then $\alpha \mapsto (T_\alpha(X), R_\alpha(X))$ is continuous on $[0, 1]$ and $\mathcal{C}^1$ on $(0, 1)$.

**Continuity/$\mathcal{C}^1$ of $\mathcal{L}$ and existence of minimizers.** Composing $(\Theta, \alpha) \mapsto (T_\alpha(X), R_\alpha(X))$ with $\mathcal{F}_\Theta$ (assumed $\mathcal{C}^1$ in $\Theta$) and $\ell$ (assumed $\mathcal{C}^1$) yields that $\mathcal{L}$ is continuous on $\mathbb{R}^p \times [0, 1]$ and $\mathcal{C}^1$ on $\mathbb{R}^p \times (0, 1)$. If $\lambda > 0$, then $\mathcal{L}(\Theta, \alpha) \to \infty$ as $\|\Theta\|_2 \to \infty$ uniformly in $\alpha$, hence a minimizer exists by the direct method/Weierstrass on the compact set $[0, 1]$ in $\alpha$. Alternatively, if $\Theta$ is constrained to a compact set, Weierstrass applies on that product set.

**Stationarity of projected first-order methods.** Consider projected gradient descent (or Adam with projection) on a compact sublevel set $\{(\Theta, \alpha) : \mathcal{L}(\Theta, \alpha) \leq c\}$, which is nonempty by existence. On

---

[1]Equivalently, $D_i(\alpha) = \sum_{k=0}^{i-1}(1-\alpha)^k = \frac{1-(1-\alpha)^i}{\alpha}$ for $\alpha \in (0, 1]$ and $D_i(0) = i$. Thus $D_i$ is continuous and strictly positive on $[0, 1]$.

such a bounded set, the gradient is locally Lipschitz (layers like linear/conv/GELU have locally Lipschitz derivatives). Standard arguments for smooth constrained nonconvex optimization then imply that every cluster point is first-order stationary (the projected gradient vanishes).

**Gradient identity and interpretability.** Let $P_\alpha(X) = (X_{\text{trend}}(\alpha), X_{\text{res}}(\alpha))$ with $X_{\text{res}}(\alpha) = X - X_{\text{trend}}(\alpha)$. By the chain rule and $\partial_\alpha X_{\text{res}} = -\partial_\alpha X_{\text{trend}}$,

$$\frac{\partial \mathcal{L}}{\partial \alpha} = \left\langle \nabla_{X_{\text{trend}}}\mathcal{L}, \ \partial_\alpha X_{\text{trend}} \right\rangle + \left\langle \nabla_{X_{\text{res}}}\mathcal{L}, \ \partial_\alpha X_{\text{res}} \right\rangle = \left\langle \nabla_{X_{\text{trend}}}\mathcal{L} - \nabla_{X_{\text{res}}}\mathcal{L}, \ \partial_\alpha X_{\text{trend}} \right\rangle,$$

which is equation 5. EMA assigns lag-$k$ weight $(1-\alpha)^k$, so the half-life $h(\alpha)$ satisfies $(1-\alpha)^{h(\alpha)} = \frac{1}{2}$, i.e., $h(\alpha) = \log 2/(-\log(1-\alpha))$ and $h(\alpha) \approx \log 2/\alpha$ for small $\alpha$. Thus smaller $\alpha$ encodes longer effective memory (smoother trend), whereas larger $\alpha$ yields a more reactive trend. The update in equation 5 aligns this memory control with the differential demand $\nabla_{X_{\text{trend}}}\mathcal{L} - \nabla_{X_{\text{res}}}\mathcal{L}$, providing an interpretable knob. $\qquad\square$

*Remark* A.1 (Boundary behavior). The map $\alpha \mapsto X_{\text{trend}}(\alpha)$ is continuous on $[0,1]$, with $X_{\text{trend}}(0)$ the running mean and $X_{\text{trend}}(1) = X$. We optimize $\alpha \in [0,1]$ via projection; all identities hold on $(0,1)$ and extend continuously to the boundary.

## A.2 PROOF OF LEMMA 1: EFFICIENCY OF HETEROGENEOUS COMPLEXITY MATCHING

*Lemma* A.2 (Efficiency of heterogeneous complexity matching). Assume the target decomposes as $f^\star = g^\star + h^\star$ with $g^\star \in \mathcal{G}_{\text{trend}}$ (low complexity) and $h^\star \in \mathcal{H}_{\text{res}}$ (high complexity). Consider a homogeneous class $\mathcal{F}_{\text{homo}}$ (e.g., a single large net) and the heterogeneous class

$$\mathcal{F}_{\text{het}} = \mathcal{F}_{\text{MLP}} + \mathcal{F}_{\text{CNN}} := \{f_1 + f_2 : \ f_1 \in \mathcal{F}_{\text{MLP}}, \ f_2 \in \mathcal{F}_{\text{CNN}}\}.$$

Assume (A1) $\mathcal{F}_{\text{het}} \subseteq \mathcal{F}_{\text{homo}}$ and (A2) $\ell$ is $L_\ell$-Lipschitz in its first argument. Then for any sample $S$ of size $n$, the uniform Rademacher generalization bound obtained with $\mathcal{F}_{\text{het}}$ is no looser than that obtained with $\mathcal{F}_{\text{homo}}$, and is strictly tighter whenever $\hat{\mathcal{R}}_S(\mathcal{F}_{\text{het}}) < \hat{\mathcal{R}}_S(\mathcal{F}_{\text{homo}})$. Moreover, when *complexity matching* holds in the sense that $\hat{\mathcal{R}}_S(\mathcal{F}_{\text{MLP}}) \approx \hat{\mathcal{R}}_S(\mathcal{G}_{\text{trend}})$ and $\hat{\mathcal{R}}_S(\mathcal{F}_{\text{CNN}}) \approx \hat{\mathcal{R}}_S(\mathcal{H}_{\text{res}})$, the bound for $\mathcal{F}_{\text{het}}$ is near-minimal among additive classes that can approximate $g^\star + h^\star$.

*Proof.* **Uniform bound.** For any function class $\mathcal{F}$ and any $\delta \in (0,1)$, with probability at least $1-\delta$ over the draw of $S$,

$$\forall f \in \mathcal{F}: \quad R_D(f) \ \leq \ \hat{R}_S(f) + 2L_\ell \hat{\mathcal{R}}_S(\mathcal{F}) + M\sqrt{\frac{\log(1/\delta)}{2n}}, \tag{8}$$

where $R_D$ is the population risk, $\hat{R}_S$ the empirical risk, $M$ bounds the loss, and the factor $L_\ell$ follows from Talagrand's contraction (see, e.g., Bartlett & Mendelson (2002); Shalev-Shwartz & Ben-David (2014)).

**Monotonicity and sub-additivity.** By set-inclusion monotonicity of Rademacher complexity and (A1),

$$\hat{\mathcal{R}}_S(\mathcal{F}_{\text{het}}) \ \leq \ \hat{\mathcal{R}}_S(\mathcal{F}_{\text{homo}}). \tag{9}$$

For any $\mathcal{A}, \mathcal{B}$, sub-additivity gives

$$\hat{\mathcal{R}}_S(\mathcal{A} + \mathcal{B}) \ \leq \ \hat{\mathcal{R}}_S(\mathcal{A}) + \hat{\mathcal{R}}_S(\mathcal{B}). \tag{10}$$

Hence

$$\hat{\mathcal{R}}_S(\mathcal{F}_{\text{het}}) = \hat{\mathcal{R}}_S(\mathcal{F}_{\text{MLP}} + \mathcal{F}_{\text{CNN}}) \leq \hat{\mathcal{R}}_S(\mathcal{F}_{\text{MLP}}) + \hat{\mathcal{R}}_S(\mathcal{F}_{\text{CNN}}). \tag{11}$$

**Comparison of bounds.** Fix any $f \in \mathcal{F}_{\text{het}}$ (hence $f \in \mathcal{F}_{\text{homo}}$ by (A1)). Applying equation 8 with $\mathcal{F} = \mathcal{F}_{\text{het}}$ and with $\mathcal{F} = \mathcal{F}_{\text{homo}}$ and using equation 9 shows that the bound via $\mathcal{F}_{\text{het}}$ is no looser, and strictly tighter if $\hat{\mathcal{R}}_S(\mathcal{F}_{\text{het}}) < \hat{\mathcal{R}}_S(\mathcal{F}_{\text{homo}})$, by a margin $2L_\ell\big(\hat{\mathcal{R}}_S(\mathcal{F}_{\text{homo}}) - \hat{\mathcal{R}}_S(\mathcal{F}_{\text{het}})\big)$.

**Effect of complexity matching.** Equation equation 11 decouples the class-level complexity penalty into two terms aligned with the target decomposition. If $\mathcal{F}_{\text{MLP}}$ and $\mathcal{F}_{\text{CNN}}$ are sized so that their complexities match $\mathcal{G}_{\text{trend}}$ and $\mathcal{H}_{\text{res}}$, then $\hat{\mathcal{R}}_S(\mathcal{F}_{\text{het}})$ is close to the minimal capacity required to represent $g^\star + h^\star$, while $\mathcal{F}_{\text{homo}}$ typically contains many functions irrelevant to this structure. This yields a uniformly smaller (or equal) complexity term in equation 8 without sacrificing approximation. $\qquad\square$

**Remark (Inductive bias).** Beyond capacity control, the additive constraint $f = f_1 + f_2$ with $f_1$ low-complexity (trend) and $f_2$ high-complexity (residual) curbs the tendency of a single high-capacity model to fit low-frequency structure, thus narrowing the set of empirical minimizers to decomposition-aligned solutions—a data-dependent benefit often not captured by class-only capacities but visible in practice.

## A.3 PROOFS OF PROPOSITIONS 3.2 AND 3.3

**Preliminaries and standing assumptions.** Throughout this section we use that the AdaFusionNet predictor $f$ is $L^*(\alpha, \mu, \text{Leak})$-Lipschitz w.r.t. the input (Prop. 3.1), i.e.,

$$\|f(x) - f(x')\| \le L^*(\alpha, \mu, \text{Leak}) \|x - x'\| \quad \text{for all } x, x' \in \mathbb{R}^d. \tag{12}$$

For the loss $\ell(\cdot, y)$ we assume $L_\ell$-Lipschitz in its first argument and, when stated, $\lambda$-smooth. These are precisely the regularity conditions used in Sec. §3. **Auxiliary lemmas.** We record two standard facts that we will use repeatedly.

**Lemma A.2** (Sub-Gaussian moment bounds). *Let $\xi \in \mathbb{R}^d$ be mean-zero sub-Gaussian with proxy variance $\sigma^2$, i.e., for every unit $u \in \mathbb{S}^{d-1}$, $u^\top \xi$ is $\sigma$-sub-Gaussian. Then*

$$\mathbb{E}\|\xi\|^2 = \text{tr}\big(\text{Cov}(\xi)\big) \le d\,\sigma^2 \quad \text{and} \quad \mathbb{E}\|\xi\| \le \sqrt{\mathbb{E}\|\xi\|^2} \le \sigma\sqrt{d}.$$

*Proof.* By definition of proxy variance, $\text{Cov}(\xi) \preceq \sigma^2 I_d$; hence $\text{tr}(\text{Cov}(\xi)) \le d\sigma^2$. Jensen's inequality yields $\mathbb{E}\|\xi\| \le \sqrt{\mathbb{E}\|\xi\|^2}$. $\qquad\square$

**Lemma A.3** (Masking/imputation are non-expansive). *Let $M \in \{0,1\}^d$ be a binary mask and $\mathcal{I} : \mathbb{R}^d \to \mathbb{R}^d$ be non-expansive (i.e., $\text{Lip}(\mathcal{I}) \le 1$). Then the maps $x \mapsto M \odot x$ and $x \mapsto \mathcal{I}(M \odot x)$ are 1-Lipschitz. Consequently, $\text{Lip}\big(f \circ \mathcal{I} \circ (M \odot \cdot)\big) \le \text{Lip}(f) = L^*(\alpha, \mu, \text{Leak})$.*

*Proof.* For any $x, x'$, $\|M \odot x - M \odot x'\|^2 = \sum_i M_i^2 (x_i - x_i')^2 \le \sum_i (x_i - x_i')^2 = \|x - x'\|^2$. Composition of Lipschitz maps multiplies constants. $\qquad\square$

**Proof of Proposition 3.2 (Prediction drift and risk inflation).** Let $x' = x + \xi$ with $\xi$ mean-zero sub-Gaussian (proxy variance $\sigma^2$). Using equation 12 with $x' = x + \xi$,

$$\|f(x + \xi) - f(x)\| \le L^*(\alpha, \mu, \text{Leak}) \|\xi\|. \tag{13}$$

Taking expectation (no independence assumptions beyond the definition of $x'$ are needed for this step),

$$\mathbb{E}\|f(x') - f(x)\| \le L^*(\alpha, \mu, \text{Leak}) \mathbb{E}\|\xi\| \le L^*(\alpha, \mu, \text{Leak}) \sigma\sqrt{d},$$

where the last inequality uses Lemma A.2. This proves the first claim.

For the risk inflation bound, write the risks under the pushed-forward distribution $\mathcal{D}_{x'}$ and the original $\mathcal{D}_x$ as expectations over $(x, y)$ and $\xi$:

$$\mathcal{R}_{\mathcal{D}_{x'}}(f) - \mathcal{R}_{\mathcal{D}_x}(f) = \mathbb{E}_{x,y,\xi}\big[\ell(f(x + \xi), y) - \ell(f(x), y)\big].$$

By $L_\ell$-Lipschitzness of $\ell(\cdot, y)$ and equation 13,

$$\big|\ell(f(x + \xi), y) - \ell(f(x), y)\big| \le L_\ell \|f(x + \xi) - f(x)\| \le L_\ell L^*(\alpha, \mu, \text{Leak}) \|\xi\|.$$

Taking expectations and using Lemma A.2 finishes the proof: $\big|\mathcal{R}_{\mathcal{D}_{x'}}(f) - \mathcal{R}_{\mathcal{D}_x}(f)\big| \le L_\ell L^*(\alpha, \mu, \text{Leak}) \sigma\sqrt{d}$.

*Quadratic refinement for squared loss.* Assume $\ell(z, y) = \frac{1}{2}\|z - y\|^2$ and $\|f(x)\| \le B_f$, $\|y\| \le B_y$ almost surely (bounded outputs/labels; see assumptions preceding Def. 2.3). Then

$$\ell(f(x + \xi), y) - \ell(f(x), y) = \big\langle f(x) - y, \ f(x + \xi) - f(x)\big\rangle + \tfrac{1}{2}\|f(x + \xi) - f(x)\|^2.$$

Taking absolute values and expectation, and using Cauchy–Schwarz and equation 13,

$$\big|\mathbb{E}[\ell(f(x + \xi), y) - \ell(f(x), y)]\big| \le (B_f + B_y) \mathbb{E}\|f(x + \xi) - f(x)\| + \tfrac{1}{2}\mathbb{E}\|f(x + \xi) - f(x)\|^2$$

$$\le (B_f + B_y) L^*(\alpha, \mu, \text{Leak}) \mathbb{E}\|\xi\| + \tfrac{1}{2} L^*(\alpha, \mu, \text{Leak})^2 \mathbb{E}\|\xi\|^2 \le (B_f + B_y) L^*(\alpha, \mu, \text{Leak}) \sigma\sqrt{d} + \tfrac{1}{2} L^*(\alpha, \mu, \text{Leak})^2 \sigma^2 d.$$

using Lemma A.2 in the last step. Compared to the purely linear bound, the second term captures the curvature of the squared loss and tightens the scaling in regimes where $\mathbb{E}\|\xi\|^2$ is informative. This is the "tighter quadratic refinement" mentioned beneath Prop. 3.2.

**Proof of Proposition 3.3 (Robustness to missing data).** Let $M \in \{0,1\}^d$ be a mask and $\mathcal{I}$ a non-expansive imputer. By Lemma A.3 and equation 12,

$$\mathrm{Lip}(f \circ \mathcal{I} \circ (M \odot \cdot)) \;\leq\; \mathrm{Lip}(f) \;\leq\; L^*(\alpha, \mu, \mathrm{Leak}).$$

Define the imputation error $e(x) := \mathcal{I}(M \odot x) - x$. Then

$$\big|\ell(f(\mathcal{I}(M \odot x)), y) - \ell(f(x), y)\big| \;\leq\; L_\ell \, \|f(x + e(x)) - f(x)\| \;\leq\; L_\ell \, L^*(\alpha, \mu, \mathrm{Leak}) \, \|e(x)\|,$$

where we used equation 12 with $x' = x + e(x)$. Taking expectations over $(x, y)$ and the mask/imputer randomness,

$$\begin{aligned}
\big|\mathcal{R}(f \circ \mathcal{I} \circ (M \odot \cdot)) - \mathcal{R}(f)\big| &\leq L_\ell \, L^*(\alpha, \mu, \mathrm{Leak}) \, \mathbb{E}\|e(x)\| \\
&\leq L_\ell \, L^*(\alpha, \mu, \mathrm{Leak}) \, \sqrt{\mathbb{E}\|e(x)\|^2} \;\leq\; L_\ell \, L^*(\alpha, \mu, \mathrm{Leak}) \, \sigma_\mathcal{I} \sqrt{\rho_{\mathrm{miss}} \, d}.
\end{aligned}$$

where the penultimate inequality is Jensen, and the last uses the assumed second-moment control $\mathbb{E}\|e(x)\|^2 \leq \sigma_\mathcal{I}^2 \, \rho_{\mathrm{miss}} \, d$. This proves Prop. 3.3.

## A.4 Proofs of Theorem 3.4 and Proposition 3.5

**Setup and notation.** Let $\mathcal{X}$ be a Polish metric space with metric $d_\mathcal{X}$, let $\mathcal{Y}$ be a (finite or countable) label space, and write $\mathcal{Z} := \mathcal{X} \times \mathcal{Y}$. A hypothesis $f : \mathcal{X} \to \mathbb{R}^m$ is measurable. The loss $\ell : \mathbb{R}^m \times \mathcal{Y} \to \mathbb{R}$ is assumed to be $L_\ell$-Lipschitz in its first argument *uniformly in $y$*, i.e.,

$$\big|\ell(u, y) - \ell(u', y)\big| \leq L_\ell \, \|u - u'\| \qquad \forall \, u, u' \in \mathbb{R}^m, \; \forall \, y \in \mathcal{Y},$$

for some norm $\|\cdot\|$ on $\mathbb{R}^m$. We further assume that $f$ is $L^*(\alpha, \mu, \mathrm{Leak})$-Lipschitz w.r.t. $d_\mathcal{X}$, namely

$$\|f(x) - f(x')\| \leq L^*(\alpha, \mu, \mathrm{Leak}) \, d_\mathcal{X}(x, x') \qquad \forall \, x, x' \in \mathcal{X}. \tag{14}$$

(As in the main text, $L^*(\alpha, \mu, \mathrm{Leak})$ is the Lipschitz modulus established earlier; here we only use that it upper-bounds the input–output sensitivity of $f$.)

For a reference distribution $P$ on $\mathcal{Z}$, we consider two standard discrepancy measures:

(i) The *total variation* distance $\mathrm{TV}(P, Q) := \sup_{A \subseteq \mathcal{Z}} |P(A) - Q(A)|$.

(ii) A *label-preserving* 1-Wasserstein distance on $\mathcal{Z}$ obtained by transporting only within each label. Formally, require $Q_Y = P_Y$ and define

$$W_1^{\mathrm{lab}}(Q, P) \;:=\; \mathbb{E}_{Y \sim P_Y}\Big[ W_1\big(Q_{X|Y}, P_{X|Y}\big) \Big] \;=\; \sum_{y \in \mathcal{Y}} P_Y(y) \, W_1\big(Q_{X|Y=y}, P_{X|Y=y}\big), \tag{15}$$

where $W_1$ on $\mathcal{X}$ uses ground metric $d_\mathcal{X}$. Equivalently, $W_1^{\mathrm{lab}}$ coincides with the optimal-transport cost on $\mathcal{Z}$ associated with the ground cost $c\big((x, y), (x', y')\big) = d_\mathcal{X}(x, x')$ if $y = y'$ and $c = +\infty$ otherwise.

We write $\mathbb{B}_W(P, \rho) := \{Q : Q_Y = P_Y, \; W_1^{\mathrm{lab}}(Q, P) \leq \rho\}$ for the corresponding Wasserstein ball.

We will frequently use the Kantorovich–Rubinstein (KR) duality: on any Polish metric space $(\mathsf{S}, d)$ and for any integrable $\varphi : \mathsf{S} \to \mathbb{R}$ with $\mathrm{Lip}(\varphi) \leq L$,

$$\Big|\mathbb{E}_Q[\varphi] - \mathbb{E}_P[\varphi]\Big| \;\leq\; L \, W_1(Q, P). \tag{16}$$

Moreover, $\sup_{W_1(Q, P) \leq \rho} \mathbb{E}_Q[\varphi] \leq \mathbb{E}_P[\varphi] + L\rho$ and likewise with $P, Q$ exchanged.

**A composition lemma.** Define $\phi : \mathcal{Z} \to \mathbb{R}$ by $\phi(x, y) := \ell\big(f(x), y\big)$. Then for each fixed $y$ the map $x \mapsto \phi(x, y)$ is $L_\ell \, L^*(\alpha, \mu, \mathrm{Leak})$-Lipschitz w.r.t. $d_\mathcal{X}$:

$$\big|\phi(x, y) - \phi(x', y)\big| \;=\; \big|\ell(f(x), y) - \ell(f(x'), y)\big| \;\leq\; L_\ell \, \|f(x) - f(x')\| \;\overset{\text{equation 14}}{\leq}\; L_\ell \, L^*(\alpha, \mu, \mathrm{Leak}) \, d_\mathcal{X}(x, x'). \tag{17}$$

This is the standard Lipschitz chain rule (composition preserves Lipschitzness with the product of moduli).

**Proof of Theorem 3.4.** *Claim (restated).* With the setup above and the label-preserving Wasserstein ball $\mathbb{B}_W(\widehat{\mathcal{D}}, \rho)$, we have

$$\sup_{Q \in \mathbb{B}_W(\widehat{\mathcal{D}}, \rho)} \mathbb{E}_Q\big[\ell\big(f(X), Y\big)\big] \;\le\; \mathbb{E}_{\widehat{\mathcal{D}}}\big[\ell\big(f(X), Y\big)\big] \;+\; L_\ell\, L^*(\alpha, \mu, \mathrm{Leak})\, \rho.$$

*Proof.* Let $\phi(x, y) = \ell(f(x), y)$ and note equation 17. Fix any $Q \in \mathbb{B}_W(\widehat{\mathcal{D}}, \rho)$. By definition of $\mathbb{B}_W$, $Q_Y = \widehat{\mathcal{D}}_Y$ and

$$W_1^{\mathrm{lab}}(Q, \widehat{\mathcal{D}}) = \mathbb{E}_{Y \sim \widehat{\mathcal{D}}_Y}\Big[W_1\big(Q_{X|Y}, \widehat{\mathcal{D}}_{X|Y}\big)\Big] \le \rho.$$

Condition on $Y = y$ and apply KR duality on the feature space $(\mathcal{X}, d_\mathcal{X})$ to the $L_\ell L^*$-Lipschitz function $x \mapsto \phi(x, y)$:

$$\mathbb{E}_{Q_{X|Y=y}}\big[\phi(\cdot, y)\big] \;\le\; \mathbb{E}_{\widehat{\mathcal{D}}_{X|Y=y}}\big[\phi(\cdot, y)\big] \;+\; L_\ell L^*(\alpha, \mu, \mathrm{Leak})\, W_1\big(Q_{X|Y=y}, \widehat{\mathcal{D}}_{X|Y=y}\big).$$

Average both sides over $y \sim \widehat{\mathcal{D}}_Y(= Q_Y)$ to obtain

$$\mathbb{E}_Q[\phi] = \mathbb{E}_{Y \sim Q_Y}\Big[\mathbb{E}_{X \sim Q_{X|Y}}[\phi(X, Y)]\Big]$$

$$\le \mathbb{E}_{Y \sim \widehat{\mathcal{D}}_Y}\Big[\mathbb{E}_{X \sim \widehat{\mathcal{D}}_{X|Y}}[\phi(X, Y)]\Big] \;+\; L_\ell L^*(\alpha, \mu, \mathrm{Leak})\, \mathbb{E}_{Y \sim \widehat{\mathcal{D}}_Y}\Big[W_1\big(Q_{X|Y}, \widehat{\mathcal{D}}_{X|Y}\big)\Big]$$

$$= \mathbb{E}_{\widehat{\mathcal{D}}}[\phi] \;+\; L_\ell L^*(\alpha, \mu, \mathrm{Leak})\, W_1^{\mathrm{lab}}(Q, \widehat{\mathcal{D}}) \;\le\; \mathbb{E}_{\widehat{\mathcal{D}}}[\phi] \;+\; L_\ell L^*(\alpha, \mu, \mathrm{Leak})\, \rho.$$

Taking the supremum over all $Q \in \mathbb{B}_W(\widehat{\mathcal{D}}, \rho)$ completes the proof. $\qquad\square$

*Remark* A.4 (On ground metrics). The label-preserving formulation above is equivalent to working on $\mathcal{Z}$ with the extended ground cost $c\big((x, y), (x', y')\big) = d_\mathcal{X}(x, x')$ if $y = y'$ and $+\infty$ otherwise; the dual then restricts to functions that are $L$-Lipschitz in $x$ uniformly over $y$. If one prefers a finite product metric, e.g., $d_\mathcal{Z}\big((x, y), (x', y')\big) = d_\mathcal{X}(x, x') + \lambda \mathbf{1}\{y \ne y'\}$ with large $\lambda$, the same conclusion follows verbatim as soon as $\ell(\cdot, y)$ is $L_\ell$-Lipschitz uniformly in $y$ and $f$ is $L^*$-Lipschitz.

**Proof of Proposition 3.5.** *Claim (restated).* Let $\mathcal{D}, \mathcal{D}'$ be two distributions on $\mathcal{Z}$. Then:

(i) (Total variation.) If $|\ell(f(x), y)| \le M$ almost surely, then

$$\Big|\mathbb{E}_\mathcal{D}\big[\ell(f(X), Y)\big] - \mathbb{E}_{\mathcal{D}'}\big[\ell(f(X), Y)\big]\Big| \;\le\; 2M\, \mathrm{TV}(\mathcal{D}, \mathcal{D}').$$

(ii) (Wasserstein.) Under the Lipschitz assumptions above, if $W_1^{\mathrm{lab}}(\mathcal{D}, \mathcal{D}') \le \rho$, then

$$\Big|\mathbb{E}_\mathcal{D}\big[\ell(f(X), Y)\big] - \mathbb{E}_{\mathcal{D}'}\big[\ell(f(X), Y)\big]\Big| \;\le\; L_\ell\, L^*(\alpha, \mu, \mathrm{Leak})\, \rho.$$

*Proof of (i).* Let $g(z) := \ell(f(x), y)$ so that $\|g\|_\infty \le M$. By the variational characterization of total variation,

$$\sup_{\|h\|_\infty \le 1} \Big|\mathbb{E}_\mathcal{D}[h] - \mathbb{E}_{\mathcal{D}'}[h]\Big| \;=\; 2\, \mathrm{TV}(\mathcal{D}, \mathcal{D}').$$

Apply this with $h = g/M$ to get

$$\Big|\mathbb{E}_\mathcal{D}[g] - \mathbb{E}_{\mathcal{D}'}[g]\Big| \;=\; M \Big|\mathbb{E}_\mathcal{D}[h] - \mathbb{E}_{\mathcal{D}'}[h]\Big| \;\le\; 2M\, \mathrm{TV}(\mathcal{D}, \mathcal{D}'),$$

as claimed. (Equivalently, using the Hahn–Jordan decomposition of the signed measure $\mathcal{D} - \mathcal{D}'$ yields $|\mathbb{E}_\mathcal{D}[g] - \mathbb{E}_{\mathcal{D}'}[g]| \le \int |g|\, d|\mathcal{D} - \mathcal{D}'| \le M\, |\mathcal{D} - \mathcal{D}'|(\mathcal{Z}) = 2M\, \mathrm{TV}(\mathcal{D}, \mathcal{D}')$.)

*Proof of (ii).* Write $\phi(x, y) = \ell(f(x), y)$ and note from equation 17 that for each label $y$ the map $x \mapsto \phi(x, y)$ is $L_\ell L^*$-Lipschitz on $(\mathcal{X}, d_\mathcal{X})$. Conditioning on $Y$ and invoking KR duality on $\mathcal{X}$ as in the proof of Theorem 3.4, we obtain

$$\Big|\mathbb{E}_\mathcal{D}[\phi] - \mathbb{E}_{\mathcal{D}'}[\phi]\Big| \;\le\; L_\ell L^*(\alpha, \mu, \mathrm{Leak})\, W_1^{\mathrm{lab}}(\mathcal{D}, \mathcal{D}') \;\le\; L_\ell L^*(\alpha, \mu, \mathrm{Leak})\, \rho,$$

which concludes the proof. $\qquad\square$

*Remark* A.5 (Integrability). The arguments above require $\phi = \ell \circ (f, \mathrm{id}_\mathcal{Y})$ to be integrable under the distributions considered. This is automatic if $\ell$ is bounded or has at most linear growth and $f$ is Lipschitz on a space with finite first moment (i.e., $\mathbb{E}[d_\mathcal{X}(X, x_0)] < \infty$ for some $x_0 \in \mathcal{X}$), which is the regime customary in distributionally robust risk bounds.

## A.5 PROOFS OF THEOREMS 3.6 AND 3.7

**Proof of Theorem 3.6 (fully detailed). Setup.** Let $\mathcal{Z}$ denote the data space and let $S = (Z_1, \ldots, Z_n) \in \mathcal{Z}^n$ be drawn i.i.d. from an unknown distribution $\mathcal{D}$. For $\theta \in \Theta$, a predictor $f_\theta$ induces a loss $\ell(f_\theta; z) \in [0, 1]$ for $z \in \mathcal{Z}$. Define the population risk $\mathcal{R}(f_\theta) := \mathbb{E}_{Z \sim \mathcal{D}}\big[\ell(f_\theta; Z)\big]$ and the empirical risk $\widehat{\mathcal{R}}_S(f_\theta) := \frac{1}{n} \sum_{i=1}^n \ell(f_\theta; Z_i)$. Fix a prior distribution $P$ on $\Theta$ independent of $S$, and let $Q = Q_S$ be any (data-dependent) posterior on $\Theta$ (we allow $Q$ to be an arbitrary measurable function of $S$; if $Q \not\ll P$ then $\mathrm{KL}(Q\|P) = +\infty$ and the bound below is trivial).

**Step 1: Exponential change of measure.** For any measurable $\phi : \Theta \to \mathbb{R}$ and any distributions $Q \ll P$,

$$\mathbb{E}_{\theta \sim Q}\big[\phi(\theta)\big] \leq \mathrm{KL}(Q\|P) + \log\Big(\mathbb{E}_{\theta \sim P}\big[e^{\phi(\theta)}\big]\Big). \tag{18}$$

*Justification.* Writing $r = \frac{dQ}{dP}$ and using the variational characterization of relative entropy together with Young's inequality,

$$\mathbb{E}_Q[\phi] = \mathbb{E}_P[r\,\phi] \leq \mathbb{E}_P\big[r \log r\big] + \log \mathbb{E}_P\big[e^\phi\big] = \mathrm{KL}(Q\|P) + \log \mathbb{E}_P\big[e^\phi\big].$$

**Step 2: A Hoeffding-type MGF bound.** Fix $\theta \in \Theta$. Set $X_i := \ell(f_\theta; Z_i) \in [0, 1]$ with mean $\mu := \mathbb{E}[X_i] = \mathcal{R}(f_\theta)$. By Hoeffding's lemma, for any $t \in \mathbb{R}$, $\mathbb{E}\big[e^{t(X_i - \mu)}\big] \leq \exp(t^2/8)$ because $X_i - \mu \in [-\mu,\, 1 - \mu] \subseteq [-1, 1]$ has range at most 1. Taking $t = \lambda/n$ and using independence over $i$,

$$\mathbb{E}_{S \sim \mathcal{D}^n}\Big[\exp\big(\lambda\big(\widehat{\mathcal{R}}_S(f_\theta) - \mathcal{R}(f_\theta)\big)\big)\Big] = \prod_{i=1}^n \mathbb{E}\Big[\exp\big(\tfrac{\lambda}{n}(X_i - \mu)\big)\Big] \leq \exp\big(\tfrac{\lambda^2}{8n}\big). \tag{19}$$

Equivalently,

$$\mathbb{E}_{S \sim \mathcal{D}^n}\Big[\exp\big(\lambda\big(\mathcal{R}(f_\theta) - \widehat{\mathcal{R}}_S(f_\theta)\big)\big)\Big] \leq \exp\big(\tfrac{\lambda^2}{8n}\big). \tag{20}$$

**Step 3: A high-probability control of the moment.** Define, for each sample $S$ and parameter $\theta$,

$$\phi_S(\theta) := \lambda\Big(\mathcal{R}(f_\theta) - \widehat{\mathcal{R}}_S(f_\theta)\Big).$$

Taking expectation in equation 20 with respect to $\theta \sim P$ and applying Fubini,

$$\mathbb{E}_S\Big[\mathbb{E}_{\theta \sim P}\big[e^{\phi_S(\theta)}\big]\Big] = \mathbb{E}_{\theta \sim P}\Big[\mathbb{E}_S\big[e^{\phi_S(\theta)}\big]\Big] \leq \exp\big(\tfrac{\lambda^2}{8n}\big).$$

Hence, by Markov's inequality, with probability at least $1 - \delta$ over the draw of $S$,

$$\mathbb{E}_{\theta \sim P}\big[e^{\phi_S(\theta)}\big] \leq \frac{1}{\delta}\exp\big(\tfrac{\lambda^2}{8n}\big). \tag{21}$$

**Step 4: Putting it together and optimizing $\lambda$.** Apply equation 18 with the function $\phi_S$ (note that equation 18 holds pointwise in $S$) and then use equation 21: with probability at least $1 - \delta$ over $S$,

$$\begin{aligned}
\lambda\, \mathbb{E}_{\theta \sim Q}\Big[\mathcal{R}(f_\theta) - \widehat{\mathcal{R}}_S(f_\theta)\Big] &= \mathbb{E}_{\theta \sim Q}\big[\phi_S(\theta)\big] \\
&\leq \mathrm{KL}(Q\|P) + \log\Big(\mathbb{E}_{\theta \sim P}\big[e^{\phi_S(\theta)}\big]\Big) \\
&\leq \mathrm{KL}(Q\|P) + \tfrac{\lambda^2}{8n} + \log(1/\delta).
\end{aligned}$$

Dividing by $\lambda > 0$ and rearranging,

$$\mathbb{E}_{\theta \sim Q}\big[\mathcal{R}(f_\theta)\big] \leq \mathbb{E}_{\theta \sim Q}\big[\widehat{\mathcal{R}}_S(f_\theta)\big] + \frac{\mathrm{KL}(Q\|P) + \log(1/\delta)}{\lambda} + \frac{\lambda}{8n}. \tag{22}$$

Optimizing equation 22 over $\lambda > 0$ via the AM–GM inequality (or by setting the derivative to zero) yields

$$\min_{\lambda > 0}\left\{\frac{a}{\lambda} + \frac{\lambda}{8n}\right\} = 2\sqrt{\frac{a}{8n}} = \sqrt{\frac{a}{2n}}, \qquad a := \mathrm{KL}(Q\|P) + \log(1/\delta).$$

Substituting back gives, simultaneously for all (data-dependent) posteriors $Q$,

$$\mathbb{E}_{\theta \sim Q}\big[\mathcal{R}(f_\theta)\big] \ \leq \ \mathbb{E}_{\theta \sim Q}\big[\widehat{\mathcal{R}}_S(f_\theta)\big] \ + \ \sqrt{\frac{\mathrm{KL}(Q\|P) + \log(1/\delta)}{2n}}.$$

This is exactly the stated PAC-Bayes inequality for losses in $[0,1]$.

*Remarks.* (i) The proof delivers a *single* event of probability at least $1 - \delta$ on which the bound holds for *every* posterior $Q = Q_S$ simultaneously; this uniformity follows because equation 18 is applied pointwise in $S$, before taking the high-probability step equation 21. (ii) If $\ell \in [a,b]$ almost surely, simply replace the factor 1 in Hoeffding's lemma by $(b-a)$, yielding the radius $(b-a)\sqrt{(\mathrm{KL}(Q\|P) + \log(1/\delta))/(2n)}$. $\qquad\square$

**Proof of Theorem 3.7 (fully detailed). Setup.** Let $(Z_1, \ldots, Z_n, Z_{n+1}) = (X_1, Y_1, \ldots, X_n, Y_n, X_{n+1}, Y_{n+1})$ be *exchangeable* random variables taking values in $\mathcal{X} \times \mathcal{Y}$. Fix a split of the observed sample indices into a training set $I_{\mathrm{tr}}$ and a calibration set $I_{\mathrm{cal}}$, with $|I_{\mathrm{cal}}| = m$. Let $\widehat{f}$ be any predictor trained *only* on $\{Z_i : i \in I_{\mathrm{tr}}\}$ (no calibration labels are used in training), and let $s : \mathcal{X} \times \mathcal{Y} \to \mathbb{R}$ be any nonconformity (or residual) score computed deterministically from $(x,y)$ and $\widehat{f}$ (e.g., $s(x,y) = |y - \widehat{f}(x)|$ in regression, but the argument is score-agnostic). Define calibration scores $V_i := s(X_i, Y_i)$ for $i \in I_{\mathrm{cal}}$ and the test score $V_{m+1} := s(X_{n+1}, Y_{n+1})$. For $\alpha \in (0,1)$, define the split-conformal threshold

$$\widehat{q}_{1-\alpha} \ := \ V_{(k)} \quad \text{with} \quad k := \lceil (m+1)(1-\alpha) \rceil, \tag{23}$$

where $V_{(1)} \leq \cdots \leq V_{(m)}$ are the order statistics of $\{V_i : i \in I_{\mathrm{cal}}\}$ and we adopt the convention $V_{(m+1)} \equiv +\infty$ when $k = m+1$. The split-conformal prediction set at a new feature $x$ is

$$\mathcal{S}(x) \ := \ \big\{ y \in \mathcal{Y} : s(x,y) \leq \widehat{q}_{1-\alpha} \big\}.$$

**Step 1: Exchangeability of scores (conditional on training).** By exchangeability of the data and the fact that $\widehat{f}$ depends only on $\{Z_i : i \in I_{\mathrm{tr}}\}$, the vector of scores

$$\big(V_i : i \in I_{\mathrm{cal}}\big) \ \cup \ \{V_{m+1}\}$$

is exchangeable *conditional* on the training data $\{Z_i : i \in I_{\mathrm{tr}}\}$ (and even conditional on $X_{n+1}$ if desired). Formally, for any permutation $\pi$ of the $m+1$ indices $I_{\mathrm{cal}} \cup \{n+1\}$,

$$\big(V_{\pi(1)}, \ldots, V_{\pi(m)}, V_{\pi(m+1)}\big) \ \overset{d}{=} \ (V_1, \ldots, V_m, V_{m+1}) \quad \text{conditionally on } I_{\mathrm{tr}} \text{ and the trained } \widehat{f}.$$

**Step 2: A rank-uniformity fact.** Consider the (random) rank of $V_{m+1}$ among the multiset $\{V_i : i \in I_{\mathrm{cal}}\} \cup \{V_{m+1}\}$ when ties are broken uniformly at random; call this rank $R \in \{1, \ldots, m+1\}$. By exchangeability, conditionally on the training data (and on the random tie-breaking), $R$ is *uniform* on $\{1, \ldots, m+1\}$, hence

$$\mathbb{P}\big(R \leq k \,\big|\, \text{train}\big) = \frac{k}{m+1}, \qquad k \in \{1, \ldots, m+1\}. \tag{24}$$

**Step 3: From ranks to the split threshold.** Fix any deterministic realizations $v_1, \ldots, v_m, v_{m+1} \in \mathbb{R}$ and their ranks. Let $q_k$ denote the $k$-th order statistic of $(v_i)_{i \in I_{\mathrm{cal}}}$ alone (with the convention $q_{m+1} = +\infty$). A simple monotonicity argument shows:

$$\big(\text{rank of } v_{m+1} \text{ among } v_1, \ldots, v_m, v_{m+1} \text{ is } \leq k\big) \implies v_{m+1} \leq q_k. \tag{25}$$

Indeed, if $v_{m+1}$ is among the $k$ smallest of the $m+1$ numbers, then among the $m$ calibration numbers there are at most $k-1$ values strictly less than $v_{m+1}$; thus the $k$-th smallest calibration value is at least $v_{m+1}$, i.e., $v_{m+1} \leq q_k$.

**Step 4: Coverage.** Combining equation 24 and equation 25 with $k = \lceil (m+1)(1-\alpha) \rceil$,

$$\mathbb{P}\big(V_{m+1} \leq \widehat{q}_{1-\alpha} \,\big|\, \text{train}\big) \ \geq \ \mathbb{P}\big(R \leq k \,\big|\, \text{train}\big) \ = \ \frac{k}{m+1} \ \geq \ 1 - \alpha.$$

Since the event $\{Y_{n+1} \in \mathcal{S}(X_{n+1})\}$ is exactly $\{V_{m+1} \leq \hat{q}_{1-\alpha}\}$ by definition of $\mathcal{S}(\cdot)$, taking total expectation yields the *marginal* coverage guarantee

$$\mathbb{P}\big\{\, Y_{n+1} \in \mathcal{S}(X_{n+1}) \,\big\} \;\geq\; 1 - \alpha.$$

*Remarks.* (i) The guarantee actually holds *conditionally* on the training sample (and, if desired, on $X_{n+1}$), since the argument above is conditional throughout. (ii) Ties in the scores only make the procedure weakly more conservative because we used "$\leq$" in the definition of $\mathcal{S}(x)$; if randomized tie-breaking (or randomized p-values) is used, one can upgrade the bound to an equality $\mathbb{P}\{Y_{n+1} \in \mathcal{S}(X_{n+1})\} = 1 - \alpha$. (iii) The choice $k = \lceil (m+1)(1-\alpha) \rceil$—together with the convention $V_{(m+1)} = +\infty$—ensures validity for all $\alpha \in (0,1)$; for $\alpha < 1/(m+1)$ the threshold becomes $+\infty$ and $\mathcal{S}(x) = \mathcal{Y}$, which is (trivially) valid. $\qquad\square$

## A.6    PROOF OF PROPOSITION 3.1 (CONSTRUCTIVE $L^*(\alpha, \mu, \text{Leak})$)

*Proof (fully detailed).* **Setup and notation.** We work in finite-dimensional Euclidean spaces and endow all spaces with the standard $\ell_2$ norm; operator norms $\|\cdot\|$ are spectral (induced $\ell_2 \to \ell_2$) norms. On product spaces $\mathcal{U} \oplus \mathcal{V}$ we use the norm $\|(u,v)\|^2 = \|u\|^2 + \|v\|^2$. For a map $F$, $\mathrm{Lip}(F)$ denotes its global Lipschitz constant with respect to the chosen norms. Let $P_\alpha : \mathbb{R}^d \to \mathbb{R}^d$ be the (possibly oblique) projector selecting the "trunk" component and $Q_\alpha := I - P_\alpha$ the complementary projector feeding the "residual" branch. By design of the representation (spectral normalization / Parseval tightness),

$$\|P_\alpha\| \leq 1, \qquad \|Q_\alpha\| \leq 1. \tag{26}$$

The branch maps $g$ (trunk) and $h$ (residual) are $L_{\mathrm{tr}}(\alpha)$- and $L_{\mathrm{res}}(\alpha)$-Lipschitz, respectively:

$$\mathrm{Lip}(g) = L_{\mathrm{tr}}(\alpha), \qquad \mathrm{Lip}(h) = L_{\mathrm{res}}(\alpha).$$

The *within-channel fusion* $\Phi_{\mathrm{within}}$ is linear on the direct sum of branch features, hence can be written as

$$\Phi_{\mathrm{within}}(u,v) = A_\alpha u + B_\alpha v, \tag{27}$$

with block operators $A_\alpha, B_\alpha$ satisfying the per-head spectral budget

$$\|A_\alpha\| \leq BL_w, \qquad \|B_\alpha\| \leq BL_w, \qquad \|\Phi_{\mathrm{within}}\| \leq BL_w, \tag{28}$$

where $B \geq 1$ captures structural multiplicity (e.g. number of per-channel fusers / heads) and $L_w$ is the normalization level. The *cross-channel* block $\Phi_{\mathrm{cross}}$ is $L_{\mathrm{cross}}$-Lipschitz; as is standard, we take $L_{\mathrm{cross}} = 1 + c_\times$ with $c_\times \ll 1$ and absorb it into $BL_w$ so that, from now on,

$$\mathrm{Lip}(\Phi_{\mathrm{cross}}) \leq 1 \quad \text{and all factors of } L_{\mathrm{cross}} \text{ are absorbed into } BL_w. \tag{29}$$

Finally, denote the channel-projector coherence by

$$\mu \;\triangleq\; \max_{k \neq \ell} \big\| P_k^\top P_\ell \big\|_{\mathrm{op}}, \tag{30}$$

and recall the leakage functional from Definition 2.3: there exists a data/model constant $C_{\mathrm{leak}} > 0$ such that for the residual content $s_{\mathrm{res}}$ one has

$$\|P_\alpha s_{\mathrm{res}}\|^2 \;\leq\; C_{\mathrm{leak}} \cdot \mathrm{Leak}_{\mathrm{res} \to \mathrm{tr}}(\alpha). \tag{31}$$

**Network factorization.** With the above blocks,

$$f \;=\; \Phi_{\mathrm{cross}} \circ \Phi_{\mathrm{within}} \circ (g \oplus h) \circ (P_\alpha \oplus Q_\alpha). \tag{32}$$

By submultiplicativity of Lipschitz constants under composition,

$$\mathrm{Lip}(f) \leq \mathrm{Lip}(\Phi_{\mathrm{cross}}) \, \mathrm{Lip}\Big(\Phi_{\mathrm{within}} \circ (g \oplus h) \circ (P_\alpha \oplus Q_\alpha)\Big) \;\leq\; \mathrm{Lip}\Big(\Phi_{\mathrm{within}} \circ (g \oplus h) \circ (P_\alpha \oplus Q_\alpha)\Big), \tag{33}$$

where we used equation 29 in the last inequality.

**Two-branch linear-fusion bound.** We first isolate the base (coherence- and leakage-free) contribution. We claim that, with $\Phi_{\text{within}}$ as in equation 27,

$$\text{Lip}\Big(\Phi_{\text{within}} \circ (g \oplus h) \circ (P_\alpha \oplus Q_\alpha)\Big) \ \le \ \|A_\alpha\| \, \text{Lip}(g) \, \|P_\alpha\| \ + \ \|B_\alpha\| \, \text{Lip}(h) \, \|Q_\alpha\|. \quad (34)$$

*Derivation.* For any $x, x' \in \mathbb{R}^d$,

$$\big\| \Phi_{\text{within}}\big(g(P_\alpha x), h(Q_\alpha x)\big) - \Phi_{\text{within}}\big(g(P_\alpha x'), h(Q_\alpha x')\big) \big\|$$
$$= \big\| A_\alpha\big(g(P_\alpha x) - g(P_\alpha x')\big) + B_\alpha\big(h(Q_\alpha x) - h(Q_\alpha x')\big) \big\|$$
$$\le \|A_\alpha\| \, \|g(P_\alpha x) - g(P_\alpha x')\| + \|B_\alpha\| \, \|h(Q_\alpha x) - h(Q_\alpha x')\|$$
$$\le \|A_\alpha\| \, \text{Lip}(g) \, \|P_\alpha\| \, \|x - x'\| + \|B_\alpha\| \, \text{Lip}(h) \, \|Q_\alpha\| \, \|x - x'\|,$$

which proves equation 34.

Using equation 26, equation 28 and the bound equation 34 in equation 33, we obtain the *base budget*

$$\text{Lip}(f) \ \le \ BL_w\Big(L_{\text{tr}}(\alpha) + L_{\text{res}}(\alpha)\Big). \quad (35)$$

**Amplification from projector coherence.** The analysis above treats the two branches as fully decoupled. In our architecture, the per-head fusion that produces the $\alpha$-prediction is permitted to couple residual features that *look like* the $\alpha$-subspace. Formally, let

$$H_\alpha(u, v) \ \triangleq \ A_\alpha u \ + \ \tilde{A}_\alpha\,(P_\alpha v) \ + \ B_\alpha v,$$

where the (learned) rows collected by $\tilde{A}_\alpha$ route residual features toward the $\alpha$-head (spectral normalization is applied row-wise so that $\|\tilde{A}_\alpha\| \le BL_w$, same as $A_\alpha, B_\alpha$). By definition of the coherence parameter equation 30, for every $z$ in the residual feature range one has the restricted projection bound

$$\|P_\alpha z\| \ \le \ \mu \, \|z\|. \quad (36)$$

Proceeding as in the derivation of equation 34, but retaining the $P_\alpha v$ term and applying equation 36, yields for any $(u_i, v_i)$

$$\|H_\alpha(u_1, v_1) - H_\alpha(u_2, v_2)\| \ \le \ BL_w\Big(\|u_1 - u_2\| + (1 + \mu)\|v_1 - v_2\|\Big).$$

With $u = g(P_\alpha x)$ and $v = h(Q_\alpha x)$ we conclude

$$\text{Lip}\Big(H_\alpha \circ (g \oplus h) \circ (P_\alpha \oplus Q_\alpha)\Big) \ \le \ BL_w\Big(L_{\text{tr}}(\alpha) + (1 + \mu)L_{\text{res}}(\alpha)\Big). \quad (37)$$

Since $\Phi_{\text{cross}}$ only mixes heads with unit Lipschitz budget (absorbed in $BL_w$ by equation 29), the same coefficient controls $f$:

$$\text{Lip}(f) \ \le \ BL_w\Big(L_{\text{tr}}(\alpha) + (1 + \mu)L_{\text{res}}(\alpha)\Big). \quad (38)$$

**Contribution from imperfect separation (leakage).** Beyond geometric coherence, the *data* may place genuine residual energy inside the $\alpha$-subspace, i.e. $P_\alpha s_{\text{res}} \neq 0$. Let $x = s_{\text{tr}} + s_{\text{res}}$ be the content decomposition, and consider two inputs $x, x'$ with increments $\Delta s_{\text{tr}} := s_{\text{tr}} - s'_{\text{tr}}$ and $\Delta s_{\text{res}} := s_{\text{res}} - s'_{\text{res}}$. The trunk-branch input difference includes the leaked residual term $P_\alpha \Delta s_{\text{res}}$ in addition to $P_\alpha \Delta s_{\text{tr}}$, so

$$\|g\big(P_\alpha x\big) - g\big(P_\alpha x'\big)\| \ \le \ L_{\text{tr}}(\alpha) \left( \|P_\alpha \Delta s_{\text{tr}}\| + \|P_\alpha \Delta s_{\text{res}}\| \right).$$

By the leakage definition equation 31 combined with Parseval/tightness (Def. 2.3),

$$\|P_\alpha \Delta s_{\text{res}}\| \ \le \ \sqrt{C_{\text{leak}} \text{Leak}_{\text{res}\to\text{tr}}(\alpha)} \, \|\Delta s_{\text{res}}\| \ \le \ \sqrt{C_{\text{leak}} \text{Leak}_{\text{res}\to\text{tr}}(\alpha)} \, \|x - x'\|. \quad (39)$$

This additional trunk-path variation is propagated by the head with factor at most $\|A_\alpha\| \le BL_w$, hence it contributes the additive term

$$BL_w \, L_{\text{tr}}(\alpha) \, \sqrt{C_{\text{leak}} \text{Leak}_{\text{res}\to\text{tr}}(\alpha)} \quad (40)$$

to the Lipschitz budget of $f$.

**Putting the pieces together.** Combining equation 38 (geometric coherence) with the leakage contribution equation 40, we obtain the constructive bound

$$\mathrm{Lip}(f) \ \leq \ BL_w\Big(L_{\mathrm{tr}}(\alpha) \ + \ (1+\mu)\, L_{\mathrm{res}}(\alpha) \ + \ L_{\mathrm{tr}}(\alpha)\, \sqrt{C_{\mathrm{leak}}\, \mathrm{Leak}_{\mathrm{res}\to\mathrm{tr}}(\alpha)}\Big) \ = \ L^*(\alpha, \mu, \mathrm{Leak})\,.$$

(41)

All constants from $\Phi_{\mathrm{cross}}$ are absorbed into $BL_w$ by equation 29. If $P_\alpha$ and $Q_\alpha$ are strict contractions ($< 1$), equation 41 tightens monotonically.

**Sanity checks.** (i) In the orthogonal/ideal case ($\mu = 0$ and $\mathrm{Leak}_{\mathrm{res}\to\mathrm{tr}}(\alpha) = 0$), the bound reduces to the base budget $BL_w\big(L_{\mathrm{tr}}(\alpha) + L_{\mathrm{res}}(\alpha)\big)$ from equation 35. (ii) If the residual branch is muted ($L_{\mathrm{res}}(\alpha) = 0$), only the trunk path and its leakage matter, as expected.

This completes the proof. □

## A.7 PROOF OF PROPOSITION 3.8

*Proof.* Let $K \in \mathbb{N}$, $z \in \mathbb{R}^K$, and $w = \mathrm{softmax}(z) \in \Delta^{K-1}$ with coordinates $w_i(z) = \exp(z_i)/\sum_{k=1}^K \exp(z_k)$. Fix $g \in \mathbb{R}^K$ and consider the softmax–gated linear form

$$\phi(z) \ :=\ w(z)^\top g.$$

We perturb the logits by $\varepsilon \sim \mathcal{N}(0, \sigma_g^2 I_K)$ and study the variance of $\phi(z + \varepsilon)$ conditional on $z$.

**Delta method and first–order reduction.** Since $\phi$ is smooth, its first–order Taylor expansion around $z$ reads

$$\phi(z + \varepsilon) = \phi(z) + \nabla\phi(z)^\top \varepsilon + R_2(z, \varepsilon),$$

(42)

where the remainder admits the integral form $R_2(z, \varepsilon) = \frac{1}{2}\int_0^1 (1-t)\, \varepsilon^\top \nabla^2\phi(z + t\varepsilon)\, \varepsilon\, dt$. Because $\mathbb{E}[\varepsilon] = 0$ and $\mathrm{Cov}(\varepsilon) = \sigma_g^2 I_K$, the delta method gives, as $\sigma_g \to 0$,

$$\mathrm{Var}_\varepsilon[\phi(z + \varepsilon)] = \nabla\phi(z)^\top \mathrm{Cov}(\varepsilon)\, \nabla\phi(z) + o(\sigma_g^2) = \sigma_g^2\, \|\nabla\phi(z)\|_2^2 + o(\sigma_g^2).$$

(43)

(The remainder satisfies $\mathbb{E}[R_2] = O(\sigma_g^2)$ and $\mathrm{Var}[R_2] = O(\sigma_g^4)$ under local boundedness of $\|\nabla^2\phi\|_{\mathrm{op}}$, so it is $o(\sigma_g^2)$.)

**Jacobian of softmax and the gradient $\nabla\phi(z)$.** Differentiating $w_i$ with respect to $z_j$ yields the well–known identity

$$\frac{\partial w_i}{\partial z_j} = w_i\, (\delta_{ij} - w_j), \qquad J_{\mathrm{sm}}(z) := \Big[\frac{\partial w_i}{\partial z_j}\Big]_{i,j} = \mathrm{Diag}(w) - ww^\top.$$

(44)

Hence

$$\nabla\phi(z) = J_{\mathrm{sm}}(z)^\top g = J_{\mathrm{sm}}(z)\, g = \big(g - (w^\top g)\mathbf{1}\big) \odot w,$$

(45)

where $\odot$ denotes the Hadamard product and $\mathbf{1}$ is the all–ones vector. Combining equation 43 and equation 45, we obtain the first–order (in $\sigma_g$) variance:

$$\mathrm{Var}_\varepsilon[\phi(z + \varepsilon)] = \sigma_g^2\, \|J_{\mathrm{sm}}(z)^\top g\|_2^2 + o(\sigma_g^2)$$

$$= \sigma_g^2\, \big\|w \odot \big(g - (w^\top g)\mathbf{1}\big)\big\|_2^2 + o(\sigma_g^2)$$

$$= \sigma_g^2 \sum_{i=1}^K w_i^2\big(g_i - w^\top g\big)^2 + o(\sigma_g^2).$$

(46)

**Operator–norm (dimension–free) upper bound.** The matrix $J_{\mathrm{sm}}(z)$ is symmetric with entries $(J_{\mathrm{sm}})_{ii} = w_i(1 - w_i)$ and $(J_{\mathrm{sm}})_{ij} = -w_i w_j$ for $i \neq j$. By Gershgorin's theorem, any eigenvalue $\lambda$ lies in at least one interval

$$\lambda \in \Big[\, (J_{\mathrm{sm}})_{ii} - \sum_{j \neq i} |(J_{\mathrm{sm}})_{ij}|\,,\ (J_{\mathrm{sm}})_{ii} + \sum_{j \neq i} |(J_{\mathrm{sm}})_{ij}|\Big] = \big[\, 0\,,\, 2w_i(1 - w_i)\,\big],$$

so $\lambda_{\max}(J_{\mathrm{sm}}(z)) \leq \max_i 2w_i(1 - w_i) \leq \frac{1}{2}$. (The constant $\frac{1}{2}$ is tight, e.g., when $K = 2$ and $w = (\frac{1}{2}, \frac{1}{2})$.) Therefore,

$$\mathrm{Var}_\varepsilon[\phi(z + \varepsilon)] \ \leq\ \sigma_g^2\, \|J_{\mathrm{sm}}(z)\|_2^2\, \|g\|_2^2 + o(\sigma_g^2) \ \leq\ \frac{\sigma_g^2}{4}\, \|g\|_2^2 + o(\sigma_g^2).$$

(47)

**Data–dependent upper bound via Loewner order.** Since $J_{\mathrm{sm}}(z) \succeq 0$ and $\|J_{\mathrm{sm}}(z)\|_2 \leq \frac{1}{2}$, diagonalizing $J_{\mathrm{sm}}(z) = Q\Lambda Q^\top$ shows $J_{\mathrm{sm}}(z)^2 = Q\Lambda^2 Q^\top \preceq \|J_{\mathrm{sm}}(z)\|_2 \, Q\Lambda Q^\top \preceq \frac{1}{2} J_{\mathrm{sm}}(z)$. Hence

$$\mathrm{Var}_\varepsilon[\phi(z+\varepsilon)] = \sigma_g^2 \, g^\top J_{\mathrm{sm}}(z)^2 g + o(\sigma_g^2) \ \leq \ \frac{\sigma_g^2}{2} \, g^\top J_{\mathrm{sm}}(z)g + o(\sigma_g^2). \tag{48}$$

Noting the identity $g^\top J_{\mathrm{sm}}(z)g = \sum_{i=1}^K w_i g_i^2 - \left(\sum_{i=1}^K w_i g_i\right)^2 = \mathrm{Var}_{i\sim w}[g_i]$, we obtain the sharper, data–dependent estimate

$$\mathrm{Var}_\varepsilon[\phi(z+\varepsilon)] \ \leq \ \frac{\sigma_g^2}{2} \mathrm{Var}_{i\sim w}[g_i] + o(\sigma_g^2). \tag{49}$$

Equations equation 46, equation 47, and equation 49 together yield

$$\mathrm{Var}_\varepsilon[\phi(z+\varepsilon)] = \sigma_g^2 \, \|J_{\mathrm{sm}}(z)^\top g\|_2^2 + o(\sigma_g^2) \ \leq \ \frac{\sigma_g^2}{4} \|g\|_2^2 + o(\sigma_g^2),$$

as claimed. $\qquad\square$

**Remark (non–isotropic noise).** If $\varepsilon \sim \mathcal{N}(0, \Sigma)$ with general $\Sigma \succeq 0$, then $\mathrm{Var}_\varepsilon[\phi(z+\varepsilon)] = \nabla\phi(z)^\top \Sigma \nabla\phi(z) + o(\|\Sigma\|)$. Consequently, $\mathrm{Var}_\varepsilon[\phi(z+\varepsilon)] \leq \lambda_{\max}(\Sigma) \|J_{\mathrm{sm}}(z)^\top g\|_2^2 + o(\|\Sigma\|)$, and the bounds above hold with $\sigma_g^2$ replaced by $\lambda_{\max}(\Sigma)$.

# B EXPERIMENTAL DETAILS

## B.1 DATASET STATISTICS

Table 5 reports the dimensionality (# variates), total length, sampling frequency, and the exact train/validation/test splits used in all experiments. For **ETT** (ETTh1/ETTh2/ETTm1/ETTm2) we follow the common 6:2:2 split; for the other datasets we follow 7:1:2. In all cases, the split counts sum exactly to the total number of time steps.[2]

Table 5: Statistics of the benchmark datasets.

| Dataset | # Variates (C) | Timesteps | Frequency | Split (Tr/Val/Te) |
|---|---|---|---|---|
| ETTh1 | 7 | 17,420 | 1 hour | 10,460 / 3,488 / 3,472 |
| ETTh2 | 7 | 17,420 | 1 hour | 10,460 / 3,488 / 3,472 |
| ETTm1 | 7 | 69,680 | 15 minutes | 41,804 / 13,936 / 13,940 |
| ETTm2 | 7 | 69,680 | 15 minutes | 41,804 / 13,936 / 13,940 |
| Weather | 21 | 52,696 | 10 minutes | 36,885 / 5,270 / 10,541 |
| Traffic | 862 | 17,544 | 1 hour | 12,279 / 1,755 / 3,510 |
| Electricity | 321 | 26,304 | 1 hour | 18,411 / 2,631 / 5,262 |
| Exchange-Rate | 8 | 7,588 | 1 day | 5,310 / 759 / 1,519 |

## B.2 IMPLEMENTATION DETAILS AND HYPERPARAMETERS

All models are implemented in `PyTorch` 1.12.1 and trained on a single NVIDIA A6000 (48 GB). We use AdamW with weight decay 0.01. The learning rate follows a cosine schedule with 5 warmup epochs; unless otherwise noted, the base LR is $5 \times 10^{-4}$ (dataset-specific deviations are listed in Table 6). Training runs for at most 50 epochs with early stopping (patience 5) based on validation loss. The look-back window $L$ is chosen per dataset from $\{96, 192, 336, 512, 720\}$. We set the label length to 48 across datasets. Following the main text, we apply RevIN before/after the core model to stabilize training and restore scale (statistics computed on the training split).

Table 6 details the final hyperparameters for ADAFUSIONNET on each dataset. Here, *Patch Length/Stride* refer to the residual stream's patching scheme; *Channel Mix Ratio* is the expansion ratio of the cross-channel MLP in the fusion block; *Initial $\alpha$* is the starting value for the learnable EMA smoothing.

---

[2]"Exchange" is the standard Exchange-Rate dataset.

Table 6: Detailed hyperparameter settings for AdaFusionNet on each dataset.

| Hyperparameter | ETTh1 | ETTh2 | ETTm1 | ETTm2 | Weather | Traffic | Electricity | Exchange |
|---|---|---|---|---|---|---|---|---|
| Look-back Window ($L$) | 336 | 336 | 336 | 336 | 336 | 192 | 336 | 96 |
| Label Length | 48 | 48 | 48 | 48 | 48 | 48 | 48 | 48 |
| Patch Length | 16 | 16 | 24 | 16 | 16 | 24 | 24 | 8 |
| Stride | 8 | 8 | 12 | 8 | 8 | 12 | 12 | 4 |
| Batch Size | 128 | 128 | 128 | 128 | 128 | 32 | 32 | 128 |
| Learning Rate | 5e-4 | 5e-4 | 6e-4 | 5e-4 | 5e-4 | 5e-4 | 5e-4 | 7e-4 |
| Dropout Rate | 0.1 | 0.12 | 0.15 | 0.2 | 0.1 | 0.2 | 0.15 | 0.2 |
| Channel Mix Ratio | 2 | 3 | 2 | 2 | 2 | 3 | 2 | 3 |
| Initial $\alpha$ | 0.2 | 0.2 | 0.2 | 0.2 | 0.2 | 0.2 | 0.2 | 0.5 |

Table 7: Weather/Traffic/Electricity/Exchange vs modern baselines (MSE/MAE).

| Dataset | Len | Metric | Ours | PatchTST | TimesNet | MICN | DLinear | RLinear |
|---|---|---|---|---|---|---|---|---|
| Weather | 96 | MSE | **0.154** | 0.207 | _0.206_ | 0.214 | 0.211 | 0.219 |
| | | MAE | **0.188** | 0.249 | _0.245_ | 0.252 | 0.250 | 0.261 |
| | 192 | MSE | **0.184** | _0.237_ | _0.237_ | 0.248 | 0.246 | 0.248 |
| | | MAE | **0.223** | 0.281 | _0.280_ | 0.284 | 0.286 | 0.285 |
| | 336 | MSE | **0.233** | 0.278 | _0.267_ | 0.277 | 0.270 | 0.275 |
| | | MAE | **0.261** | 0.320 | _0.317_ | 0.323 | 0.318 | 0.326 |
| | 720 | MSE | **0.314** | 0.334 | _0.333_ | 0.344 | 0.340 | 0.342 |
| | | MAE | **0.318** | _0.366_ | _0.366_ | 0.370 | 0.367 | 0.367 |
| Traffic | 96 | MSE | 0.471 | _0.450_ | **0.443** | 0.463 | 0.456 | 0.459 |
| | | MAE | 0.267 | _0.259_ | _0.257_ | 0.262 | **0.253** | 0.264 |
| | 192 | MSE | **0.464** | 0.471 | 0.467 | 0.467 | 0.469 | _0.465_ |
| | | MAE | **0.264** | 0.273 | _0.265_ | 0.267 | 0.268 | _0.267_ |
| | 336 | MSE | **0.475** | _0.479_ | 0.480 | 0.490 | 0.485 | 0.484 |
| | | MAE | **0.287** | 0.305 | _0.295_ | 0.298 | 0.297 | 0.299 |
| | 720 | MSE | _0.505_ | **0.475** | 0.520 | 0.529 | 0.527 | 0.524 |
| | | MAE | 0.323 | **0.299** | 0.317 | 0.315 | _0.312_ | 0.320 |
| Electricity | 96 | MSE | **0.145** | _0.215_ | 0.252 | 0.256 | 0.255 | 0.259 |
| | | MAE | **0.241** | _0.301_ | 0.314 | 0.320 | 0.309 | 0.319 |
| | 192 | MSE | **0.166** | _0.233_ | 0.261 | 0.266 | 0.267 | 0.272 |
| | | MAE | **0.261** | _0.307_ | 0.324 | 0.330 | 0.326 | 0.332 |
| | 336 | MSE | **0.174** | _0.236_ | 0.268 | 0.274 | 0.274 | 0.278 |
| | | MAE | **0.267** | _0.312_ | 0.330 | 0.336 | 0.334 | 0.336 |
| | 720 | MSE | **0.198** | _0.256_ | 0.302 | 0.309 | 0.309 | 0.315 |
| | | MAE | **0.291** | _0.330_ | 0.360 | 0.361 | 0.359 | 0.367 |
| Exchange | 96 | MSE | **0.084** | _0.167_ | 0.191 | 0.203 | 0.196 | 0.203 |
| | | MAE | **0.196** | _0.259_ | 0.267 | 0.275 | 0.274 | 0.280 |
| | 192 | MSE | **0.180** | 0.280 | _0.279_ | 0.287 | 0.284 | 0.287 |
| | | MAE | **0.299** | _0.334_ | 0.335 | 0.340 | 0.336 | 0.340 |
| | 336 | MSE | 0.405 | **0.388** | 0.392 | 0.397 | _0.389_ | 0.400 |
| | | MAE | 0.459 | 0.414 | **0.405** | 0.419 | _0.412_ | 0.420 |
| | 720 | MSE | **0.724** | 0.776 | 0.779 | 0.780 | _0.768_ | 0.781 |
| | | MAE | **0.662** | 0.668 | 0.676 | _0.664_ | 0.668 | 0.667 |

## B.3 FULL RESULTS ON WEATHER/TRAFFIC/ELECTRICITY/EXCHANGE (MOVED FROM THE MAIN TEXT)

## C ADDITIONAL DIAGNOSTICS AND VISUALIZATIONS

This appendix augments the main paper with ten diagnostic figures that clarify *why* AdaFusionNet achieves strong—often SOTA—accuracy across eight public benchmarks and four horizons (96, 192, 336, 720). The visuals progress from phenomenon to mechanism to outcome. First (Figs. 3–5), we visualize **trend contamination** (spectral leakage of high-frequency dynamics into the learned trend) when composite signals are processed homogeneously. Next (Figs. 6–7), ablations demonstrate that making the exponential moving average (EMA) *learnable* is critical to accuracy. We then

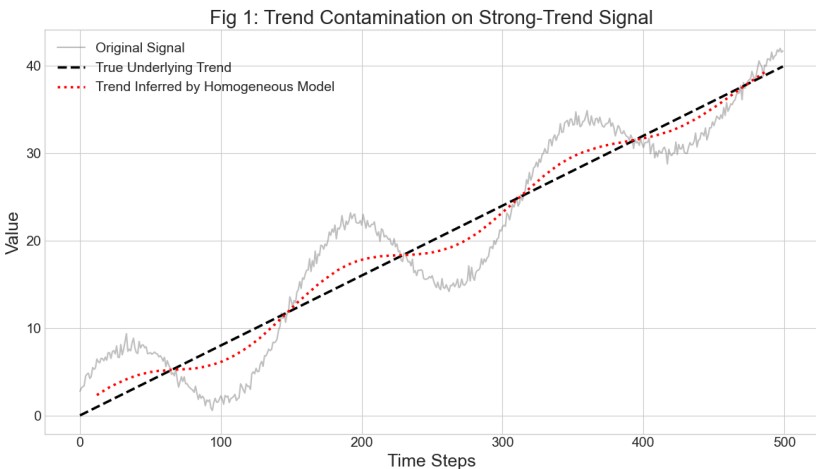

Figure 3: **Trend contamination on a strong-trend signal.** Grey: composite series; black dashed: ground-truth trend; red dotted: trend inferred by a homogeneous model trained on the raw series. Even with a dominant global drift, the inferred "trend" exhibits seasonal oscillations—a direct visualization of spectral leakage into the low-pass component that hampers long-horizon extrapolation.

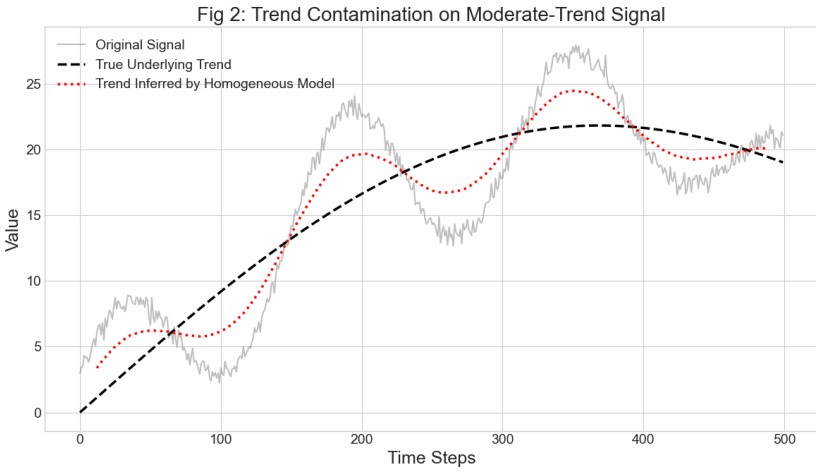

Figure 4: **Trend contamination on a moderate-trend signal.** Under moderate drift, the homogeneous model again absorbs periodic structure into the trend proxy, foretelling biased long-range forecasts.

establish interpretability of the learned smoothing (Figs. 8–9), before contrasting *heterogeneous specialization* against homogeneous variants (Fig. 10). Finally (Figs. 11–12), we examine internal representations, showing that AdaFusionNet yields clean trend/residual features aligned with our theory. These diagnostics directly support the main tables, where AdaFusionNet is best or second-best in the vast majority of settings, with the largest gains at long horizons.

**Relation to main tables and breadth of evidence.** The above diagnostics align with the extensive quantitative results summarized in Tables 1–4 and the ablation in Table 5 of the main text: (i) on *ETT*, AdaFusionNet is best across all horizons on ETTh2, ETTm1, and ETTm2 (e.g., ETTh2 at H=720: MSE 0.384 vs. iTransformer 0.471; ETTm2 at H=96: MSE 0.160 vs. 0.213), and consistently second-best on ETTh1, often matching the leader on MAE; (ii) on *Weather* and *Electricity*, AdaFusionNet leads at all horizons (e.g., Electricity MSE 0.145/0.166/0.174/0.198 at H=96/192/336/720); (iii) on *Exchange-Rate*, we dominate at H=96, 192, and 720 and remain competitive at H=336; (iv) on *Traffic*, our model remains competitive on this high-dimensional, noisy

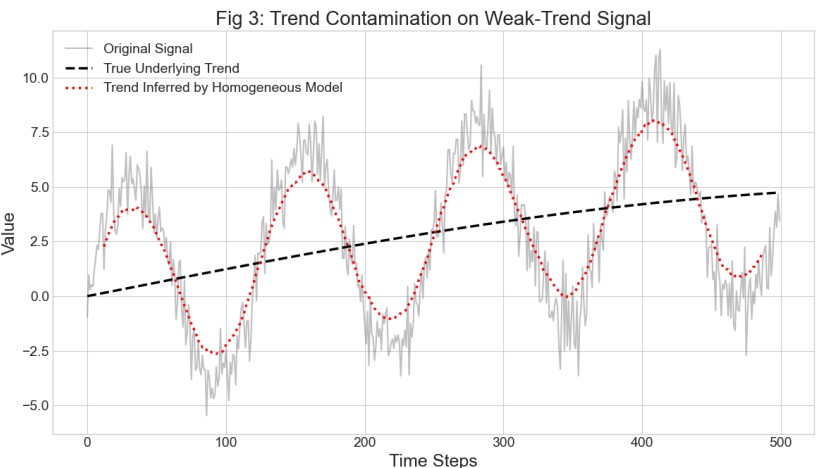

Figure 5: **Trend contamination on a weak-trend signal.** When the global drift is weak, contamination becomes severe: seasonal ripples dominate the inferred trend, blurring the intended separation of slow vs. fast dynamics. This motivates AdaFusionNet's *disentangle → specialize → fuse* pipeline.

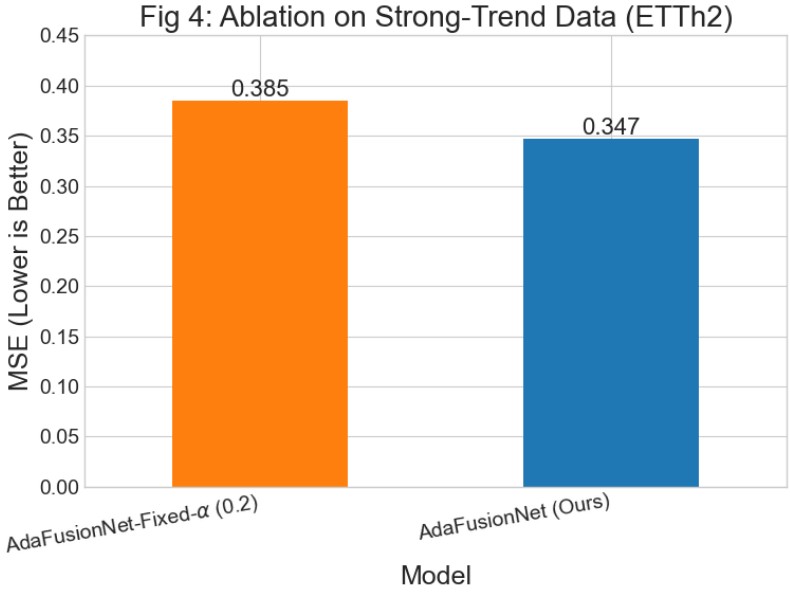

Figure 6: **Ablation on ETTh2 (prediction length 192).** Replacing the learnable EMA with a fixed smoothing factor $\alpha = 0.2$ degrades accuracy: MSE increases from **0.347** to $0.385$ (and MAE from $0.380$ to $0.401$ in the main text), demonstrating that *adaptive* disentanglement measurably reduces spectral leakage and improves long-horizon quality.

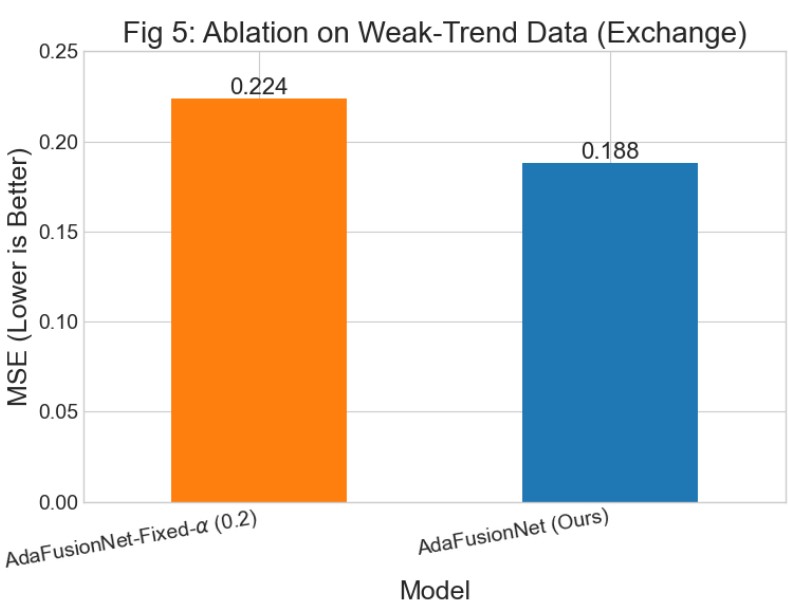

Figure 7: **Ablation on Exchange-Rate (prediction length 192).** The same trend holds in weak/volatile regimes: a fixed $\alpha$ raises MSE from **0.188** to $0.224$ (and MAE from $0.311$ to $0.345$ in the main text), confirming that learning $\hat{\alpha}$ is beneficial across dynamics.

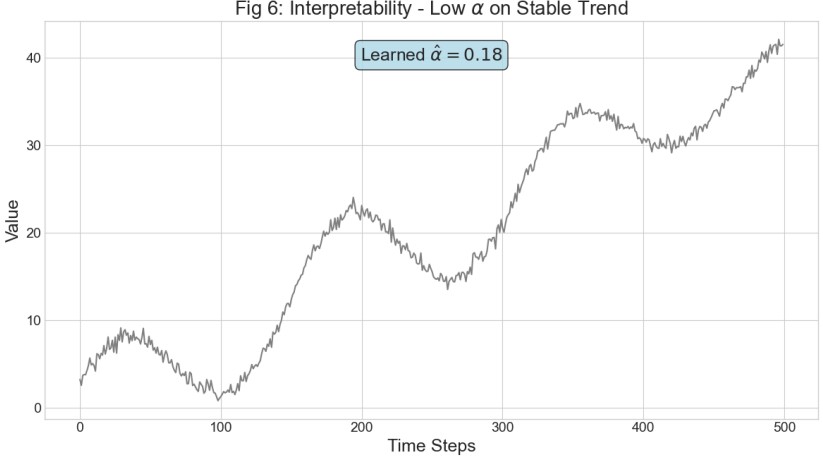

Figure 8: **Interpretability: learned smoothing on a stable trend.** For stable long-range structure, the model converges to a small $\hat{\alpha}$, implying a long EMA half-life $h(\alpha) = \log 2/[-\log(1 - \alpha)]$; most variation is routed to the trend stream, supporting reliable extrapolation.

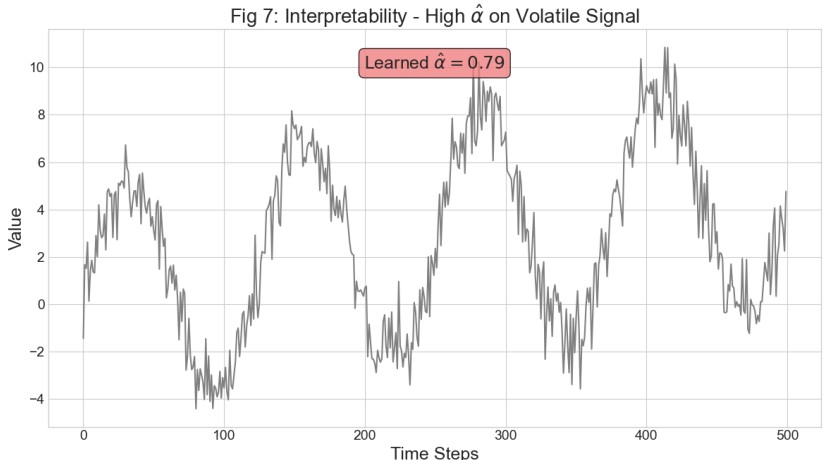

Figure 9: **Interpretability: learned smoothing on a volatile signal.** For highly volatile series, the model chooses a large $\hat{\alpha}$ (short half-life), keeping the trend reactive while delegating fast fluctuations to the residual stream—behavior predicted by the adaptive-projection gradient identity.

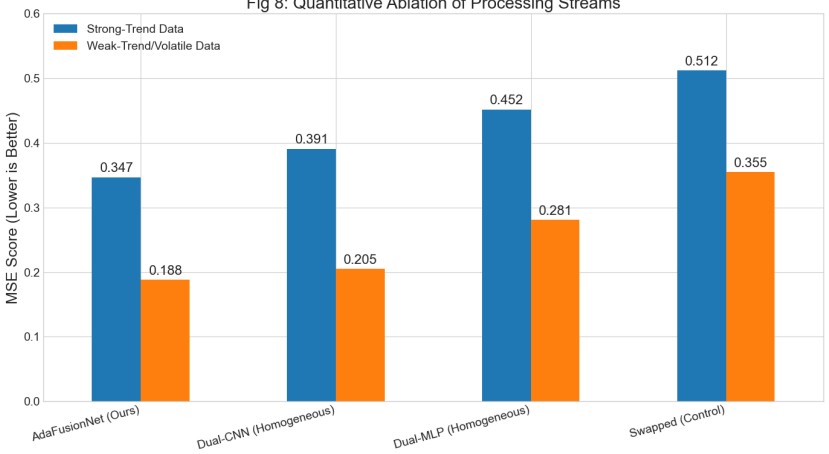

Figure 10: **Quantitative ablation of processing streams.** AdaFusionNet (MLP for trends; patch-wise CNN for residuals) outperforms Dual-CNN, Dual-MLP, and a swapped-control across strong-trend and weak/volatile settings, corroborating the *complexity matching* argument behind heterogeneous specialization.

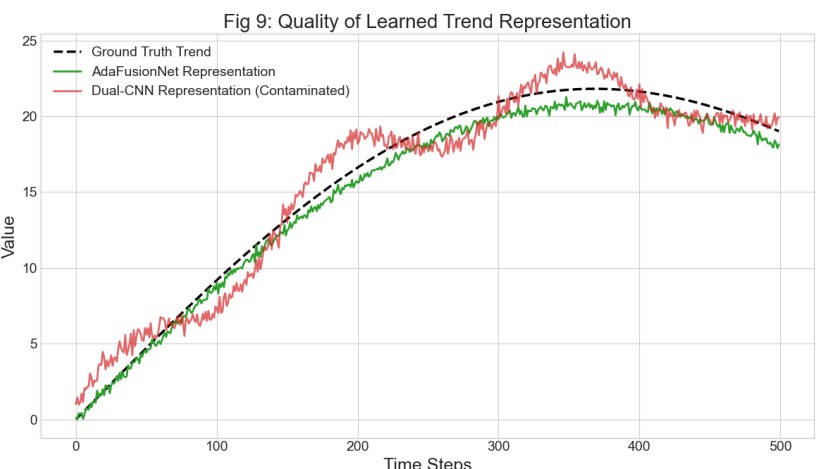

Figure 11: **Quality of the learned trend representation.** Our trend representation (green, compared to the black dashed ground truth) remains smooth and aligned with global structure, whereas a homogeneous Dual-CNN baseline (red) exhibits pronounced seasonal ripples—evidence of contamination that explains inferior long-range forecasts.

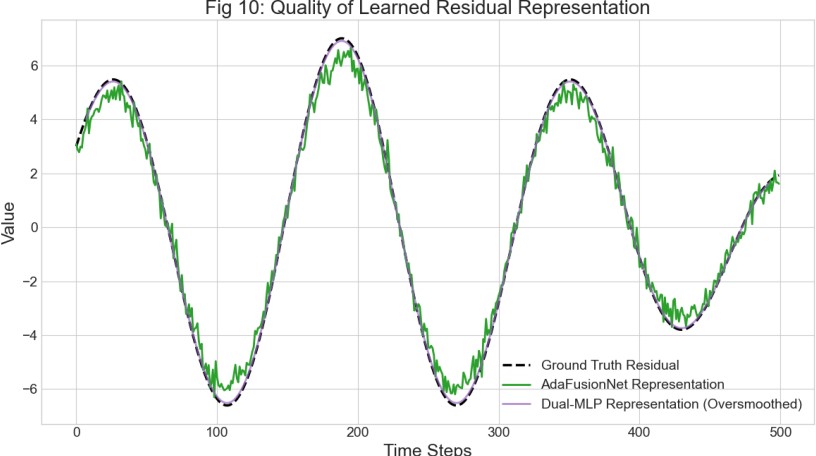

Figure 12: **Quality of the learned residual representation.** Our residual stream preserves sharp high-frequency dynamics and local shape, while a homogeneous Dual-MLP oversmooths and loses critical short-term information. Together with Fig. 11, this shows that disentangling then specializing yields cleaner internals and stronger long-horizon accuracy.

benchmark. Across datasets and horizons, gains are larger at longer horizons, consistent with our claim that reducing trend contamination benefits long-range forecasting. The ablation with fixed $\alpha$ (Table 5) quantitatively isolates the effect of *adaptive decomposition*, and the stream ablations (Dual-MLP/CNN, swapped control) support *heterogeneous specialization*. All figures in this appendix are produced with the same training protocol, splits, and hyperparameters reported in Appendix B (dataset statistics and per-dataset settings), ensuring strict comparability with the main tables.

