# OpenReview forum: "AdaFusionNet: Disentangling and Fusing Asynchronous Patterns for Long-Range Time Series Forecasting"
_ICLR.cc/2026/Conference — ICLR 2026 Conference Withdrawn Submission_

### Official Review · Reviewer_vwVe · 2025-10-30

**Soundness:** 1
**Presentation:** 1
**Contribution:** 2
**Rating:** 2
**Confidence:** 3

**Summary:**

This paper proposes AdaFusionNet, a disentangle-specialize-fuse pipeline for time series forecasting. It includes an EMA to extract the trend and MLP/CNN fusion to yield predictions. It reports strong performance on a few datasets.

**Strengths:**

S1 (performance): The proposed method reports strong empirical performance across a few datasets.

S2 (simplicity): The proposed modules are simple and easy to understand.

S3 (illustration): The paper provides an illustrative figure (Figure 2).

**Weaknesses:**

W1 (poor presentation): The proposed method is introduced in only 20 lines of text with almost no explanations. The majority of the text is jabberwocky as I will explain in W2-5.

W2 (no proofs): Theorems 2.4--2.10 have no proofs.

W3 (inexistent theorems): The appendix provides the proofs of "Theorem 1" and "Lemma 1", but Theorem 1 and Lemma 1 do not even appear in the paper.

W4 (trivial/wrong theorems in Section 2): Section 2 includes a lot of theoretical results, but all of them are either (i) extremely trivial or (ii) wrong. For example, (i) Proposition 2.7 is just a simple application of the chain rule in calculus; (ii) many terms in Theorem 2.4 are undefined.

W5 (irrelevant theorems in Section 3): Theoretical results in Section 3 are just general learning theory irrelevant to the proposed method. This paper has not attempted to connect these general theories to their work. This paper even fails to explain what these theories mean.

**Questions:**

See weaknesses.

**Details Of Ethics Concerns:**

Considering the extreme inconsistency between the main body and the appendix, it seems that the majority of this paper is generated by hallucinated LLMs. However, such excessive LLM usage is not declared in the paper.

---

### Official Review · Reviewer_zDV3 · 2025-10-31

**Soundness:** 2
**Presentation:** 2
**Contribution:** 2
**Rating:** 2
**Confidence:** 3

**Summary:**

### *Summary*

- This work proposes AdaFusionNet, an architecture that forecasts long-horizon time series. AdaFusionNet learns a low-pass filter to disentangle trend and residual streams, then processes the trend and residual streams separately, and fuses them back together at the end of the architecture to obtain a final prediction. The authors provide multiple theoretical results related to robustness of the proposed model. The authors perform experiments on 8 common long-horizon time series forecasting datasets

### *Contributions*

- AdaFusionNet, an architecture with two key contributions:
  - A learnable decomposition module
  - Separate processing of trend and residuals by MLPs and CNNs
- A theoretical analysis under Lipschitz-style constraints

**Strengths:**

### *Originality*

- A novel decomposition-based architecture with strong performance on known datasets

### *Quality*

- Ablations of key architectural contributions

### *Clarity*

- Succinct writing, well structured
- Clearly laid-out contributions

### *Significance*

- Furthers the discussion around decomposition-based time series forecasting methods

**Weaknesses:**

### *Originality*

- Baselines and related work is missing for 2024-2025. A simple search turned these up:
  - Hao, Jianhua, and Fangai Liu. "Improving long-term multivariate time series forecasting with a seasonal-trend decomposition-based 2-dimensional temporal convolution dense network." Scientific Reports 14.1 (2024): 1689\.
  - Piao et al. (2024), “Fredformer: Frequency Debiased Transformer for Time Series Forecasting” (KDD 2024\)
  - Tian et al. (2024), “MultiWaveNet: A long time series forecasting framework based on multi-scale analysis and multi-channel feature fusion.”

### *Quality*

- These datasets are saturated. See [https://arxiv.org/abs/2510.02729](https://arxiv.org/abs/2510.02729).
- **Dated baselines**: TimeXer tops the thuml library, but is missing from your tables. There is no comparison to SOTA models, and all related works go back to 2023 whereas it’s nearly 2026\. I would check out [https://huggingface.co/spaces/Salesforce/GIFT-Eval](https://huggingface.co/spaces/Salesforce/GIFT-Eval) for a list of relevant recent baseline methods.
- There is **insufficient analysis** to corroborate the **why it works** section starting on L481 because the experimental protocol is missing details. What was your hyperparameter selection strategy for testing these different architectures? Beyond demonstrating the results are better through ablations, there needs to be more investigation of the key components. For example, have you tried values other than \\alpha=0.2 (L414)?  I would start by varying the learning rate for the different architectural possibilities/ablations and seeing how the results change. If your architecture holds up, this additionally provides defense against claims of hyperparameter sensitivity.
- Results do not line up with previous work (e.g. TimeMixer, table 3\)
- No analysis of FLOPs/runtimes
- You may want to consider bootstrapping confidence intervals to validate whether your model is statistically significantly better than the baselines
- The results for the ablation on Heterogeneous processing can just go in the appendix instead of being omitted completely.
- Ideally, there’d be some sort of analysis of the internal representations to validate whether your architecture is actually resulting in cleaner internal representations.

### *Clarity*

- Figure fonts are tiny, which makes them illegible
- The theoretical text in section 3 is so dense as to be hard to parse. This makes the main takeaways unclear.
  - The phrase “Fusion weights …” starting on L115 is not even a coherent sentence.
  - Domain-specific acronyms are not introduced “e.g. PSD”.
  - Backloading the discussion of the theorems is a good space-saving technique, but is not so good for reading the paper

**Questions:**

The main items holding my score back are:

- A more recent, comprehensive set of baselines
- Evaluation on a non-saturated benchmark
- Improved empirical analysis accompanying the **why it works** section (L481)

Each of those are major points, each worth about two points in my mind. Completely all of them would raise my score to accept. However, without all three of them, I don't think I can go above a 4.

---

### Official Review · Reviewer_ybdr · 2025-10-31

**Soundness:** 1
**Presentation:** 2
**Contribution:** 2
**Rating:** 0
**Confidence:** 5

**Summary:**

The paper addresses the overpopulated long-horizon forecasting task and proposes AdaFusionNet, an architecture designed to mitigate “spectral leakage” by disentangling, specializing, and fusing forecast components.

However, the model is tested only on minuscule/over-exploited long-horizon benchmarks, casting doubts on the validity of their findings and generalization to real applications.

The theoretical foundation of AdaFusionNet relies on stationarity assumptions, which conflict with the presence of trends (a central challenge in long-horizon forecasting).

**Strengths:**

- The authors present a reasonable method of forecast decomposition. However, the novelty of their approach is quite limited. See NBEATS/NHITS projections.

- The authors should read the ESRNN paper. ESRNN has a very similar intuition to that of AdaFusionNet. The overlap is uncanny.

**Weaknesses:**

- AdaFusionNet novelty is quite limited. It is basically rediscovering the M4 competition winner method, the ESRNN. It is available for free in the International Journal of Forecasting portal.

- The authors make bold claims about the ability of the method to recover uncontaminated trends. Perhaps they should demonstrate this ability with simulated data, where they can control the trend signal. It is a wasted opportunity to limit your analysis to the long-horizon datasets.

- There are several papers in the statistical and machine learning forecasting literature that deal with trend decomposition. Please update your literature to mention relevant literature.

- The experiments in the paper are limited to ETT, Exchange, ILI Traffic, and Weather, which at this point are analogous to MNIST in forecasting research. Authors should strive to enhance the comprehensiveness of the datasets on which they base their ideas. Perhaps M1, M3, and M4 can help to increase confidence in the validity of their approach.

- From the hundreds of papers on the long-horizon forecasting task, why are the authors restricting their comparison to iTransformer, ETSFormer, CARD, TimeMixer, AutoFormer, Informer, and FEDFormer? None of these methods is "state-of-the-art".

- The authors omit any mention of AdaFusionNet hyperparameters. The authors also omit the code. There are several replicability problems.

**Questions:**

- Why is Exponential Moving Average (EMA) not introduced at the beginning or in the abstract?

- Theorem 2.6 relies on stationarity. Time series more often than not are not stationary. What is the applicability of this theorem?

- It has been shown that for most long-horizon datasets using multivariate series is not reasonable. Why is the architecture multivariate instead of univariate?

---

### Official Review · Reviewer_ezhF · 2025-11-01

**Soundness:** 3
**Presentation:** 3
**Contribution:** 4
**Rating:** 6
**Confidence:** 3

**Summary:**

This paper addresses the trend contamination problem in long-horizon time series forecasting, where high-frequency fluctuations leak into learned trends, impairing extrapolation accuracy. The model follows a disentangle-specialize-fuse paradigm: (1) a learnable exponential moving average low-pass filter adaptively separates trend and residual components via a trainable smoothing parameter; (2) heterogeneous streams match complexity to each component -- a lightweight MLP for the smooth trend and a patch-wise CNN for volatile residuals; (3) a fusion block combines per-channel predictions and models cross-variate dependencies. Theoretically, the authors model the decomposition as an adaptive projection, derive a leakage-aware risk bound with gradient updates, and establish generalization, robustness, and uncertainty guarantees. Empirically, AdaFusionNet achieves strong performance, with larger gains at longer horizons. Ablations confirm the value of learnable α and heterogeneous streams.

**Strengths:**

S1: The authors propose a learnable decomposition method that adaptively separates trend and residual components. As shown in the visual analysis, the extracted trend contains fewer periodic features, indicating effective disentanglement from high-frequency dynamics. This decomposition approach can also serve as a general plug-in module for other models focusing on seasonal–trend disentanglement.

S2: The leakage-aware risk decomposition (Theorem 2.4) provides mechanistic explanations for why learning \alpha reduces spectral leakage. Section 3 rigorously links spectral leakage and coherence to robustness, deriving leakage-aware risk bounds that improve stability under noise, missing data, and distribution shifts. It further provides PAC-Bayes and conformal guarantees for calibrated uncertainty.

S3: The proposed framework outperforms advanced baselines. Ablation studies demonstrate the advantages of key components.

**Weaknesses:**

W1: The experiments show the effectiveness of the approach within the MLP-CNN and fuse architecture, but it remains unclear whether the proposed decomposition mechanism is universally applicable to other architectures. If it does, the approach would be even more valuable and broadly impactful.

W2: Many real-world time series contain sudden yet continuous regime changes or rapidly evolving patterns (e.g., traffic spikes and stock market). The proposed adaptive decomposition may struggle to fully capture such fast transitions, as the learnable EMA filter primarily models gradual variations in trend–residual dynamics. The current visualizations also mainly focus on relatively simple trends and periodic patterns, which may not fully reflect the model’s behavior on more complex, rapidly changing, or irregular real-world series.

W3: The use of MLP for trend modeling and CNN for residual modeling is mainly motivated by intuitive matching. However, the paper provides limited empirical or theoretical justification for this specific pairing.

**Questions:**

Q1: Can the proposed adaptive decomposition mechanism generalize to other architectures beyond the MLP–CNN design, such as Transformers or recurrent models?

Q2: How does the method perform on datasets with rapid regime changes or highly non-stationary patterns? Will the proposed framework produce over-smoothing results? Could the authors provide visualizations or analyses on such challenging scenarios?

Q3: What motivated the specific choice of MLP for trends and CNN for residuals? Have the authors tried alternative architectural pairings or verified whether this design choice is essential for the observed performance gains?

---

### Note · Authors · 2025-11-12

I have read and agree with the venue's withdrawal policy on behalf of myself and my co-authors.